# Learning from A Single Graph is All You Need for Near-Shortest Path Routing

## Abstract

We propose a simple algorithm that needs only a few data samples from a single graph for learning local routing policies that generalize across classes of geometric random graphs in Euclidean and hyperbolic metric spaces. We thus solve the all-pairs near-shortest path problem by training deep neural networks (DNNs) that let each graph node efficiently and scalably route (i.e., forward) packets by considering only the node's state and the state of the neighboring nodes. Our algorithm design exploits network domain knowledge in the selection of input features and in the selection of a "seed graph" and its data samples. The leverage of domain knowledge provides theoretical assurance that the seed graph and node subsampling suffice for learning that is generalizable, scalable, and efficient. Remarkably, one of these DNNs we train —using distance as the only input feature— learns a policy that exactly matches the well-known Greedy Forwarding policy, which forwards packets to the neighbor with the shortest distance to the destination. We also learn a new policy, which we call *Greedy Tensile* routing —using both distance and stretch factor as the input features— that almost always outperforms greedy forwarding. We demonstrate the explainability and ultra-low latency runtime operation of Greedy Tensile routing by symbolically interpreting its DNN in low-complexity terms of two linear actions.

## 1 Introduction

There has been considerable interest in machine learning to mimic the human ability of learning new concepts from just a few instances. While human learning with high data efficiency is yet to be matched by formal frameworks for machine learning, such as the language-identification-in-the-limit framework (Gold, 1967) and probably approximately correct (PAC) learning (Valiant, 1984), few-shot (or one-shot) learning methods (Muggleton, 1996; Wang et al., 2020; Muggleton, 2023) have sought to bridge the gap by using prior knowledge to rapidly generalize to new instances given training on only a small number of samples with supervised information.

In this paper, we explore the learnability of policies that generalize in the domain of graph routing, where a one-size-fits-all solution is challenging when the class of graphs has strict scalability limits for network capacity (Xue & Kumar, 2006) or has significant graph dynamics (Hekmat & Van Mieghem, 2004; Grossglauser & Tse, 2002). Manually designed algorithms and heuristics often cater to particular network conditions and come with tradeoffs of complexity, scalability, and generalizability across diverse graphs. Many machine learned algorithms in this space incur relatively high computational complexity during training (Reis et al., 2019), have high overhead at run-time which limits their use in graphs with high dynamics, or are applicable only for small-scale graphs or graphs of limited types (e.g., high density graphs). Our work focuses attention on answering the following question:

> *Can we design a high data efficient machine learning algorithm for graph routing based on local search that addresses complexity, scalability and generalizability issues all at once?*

We answer this question in the affirmative for the all-pairs near-shortest path (APNSP) problem over the class of uniform random graphs in Euclidean and in hyperbolic metric spaces. It is well known that uniform random graphs in Euclidean metric spaces can represent the topologies inherent

in wireless networks and that the graphs in hyperbolic metric spaces can represent the tree-like topologies of the Internet and social networks where node degree follows a power-law distribution (Boguná et al., 2010; Verbeek & Suri, 2014); the policies we learn are thus broadly applicable to real-world wireless and wired networks. Our key insight is that —in contrast to pure black-box approaches— domain knowledge suffices to theoretically guide the selection of "seed" graph(s) and corresponding sparse training data for efficiently learning models that generalize to (almost) all graphs in these geometric classes.

To motivate our focus on local search, we recall that approaches to solve the APNSP problem can be divided into two categories: global search and local search. Global search encodes the entire network state into graph embeddings (Narayanan et al., 2017) and finds optimal paths, whereas local search needs only node embeddings (Grover & Leskovec, 2016) to predict the next forwarder on the shortest path. The model complexity (in time and space) resulting from the latter is inherently better than the former, as is the tolerance to network perturbations. The latter can even achieve stateless design, as is illustrated by geographic routing (Cadger et al., 2012) where packet forwarding can be based on using only the location of the forwarding node and the destination. In other words, local search can achieve scalability and dynamic adaptation in a fashion that is relatively independent of the network configuration. We seek to achieve these properties by learning a low-complexity policy that by virtue of its generalizability can be instantiated and adapted in real-time.

We model the APNSP problem as a Markov decision process (MDP) and propose a DNN-based approach to learn a "single-copy" local routing policy that at each routing node and at each time only considers the states from that node and one of its neighbors to predict a local metric (a $Q$-value) for routing. Routing thus uses a single neighbor for which the $Q$-value is the largest. For achieving efficient learning that generalizes over a class of graphs, we develop a theory based on the similarity between the local ranking of node neighbors in terms of $Q$-value and the global ranking with respect to the (shortest) path length metric. If local input features can be chosen to thus achieve high similarity for most nodes in almost all graphs in the chosen class, the APNSP objective can be realized with high probability by training a DNN that characterizes the local metric of each neighbor as a potential forwarder. Moreover, the DNN policy can generalize even if it is trained from only a few data samples chosen from a single "seed" graph. The theory guides our selection of input features as well as corresponding training data and is corroborated by empirical validation of our learned routing solutions.

Our approach thus yields a light-weight solution to graph routing in chosen classes of graphs, in the sense that (a) the routing policy is rapidly learned from a small dataset that is easily collected from a single "seed" graph; (b) the learned policy can be used on all nodes of a graph, and is able to generalize across almost all graphs in the classes *without additional training on the target networks*; and (c) the routing decision only depends on the local network state, i.e., the state of the node and its one-hop neighbor nodes.

Our main contributions and findings are as follows: First, generalization from few-shot learning from a single graph is feasible for APNSP and theoretically assured by domain knowledge. Second, domain knowledge also guides the selection of input features and training samples to increase the training efficiency and testing accuracy. Third, learning from a single graph using only a distance metric matches the well-known greedy forwarding routing. Fourth, learning from a single graph using both distance and node stretch factor relative to a given origin-destination node pair yields a new policy, *Greedy Tensile* routing, that achieves even better generalized APNSP routing. Fifth, both these learned policies can be symbolically interpreted in a low complexity fashion—they are approximated by policies with one and two linear actions respectively. Lastly, reinforcement learning from a single graph achieves comparable generalization performance for ASNSP.

## 2 PROBLEM FORMULATION FOR GENERALIZED ROUTING

Consider the class $\mathbb{G}$ of all graphs $G = (V, E)$ whose nodes are uniformly randomly distributed over a 2-dimensional geometric space, that is either an Euclidean space or a hyperbolic space. Each node $v \in V$ knows its global coordinates. For any pair of nodes $v, u \in V$, edge $(v, u) \in E$ holds if and only if the distance between $v$ and $u$ is at most the communication radius $R$.

For the case of $G$ in a 2-dimensional Euclidean plane, we let $R$ be a user-defined constant. Let $\rho$ denote the network density, where network density is defined to be the average number of nodes per

$R^2$ area, and $n$ the number of nodes in $V$. It follows that all nodes in $V$ are distributed in a square whose side is of length $\sqrt{\frac{n \times R^2}{\rho}}$.

For the case of $G$ in a 2-dimensional hyperbolic plane, all nodes in $V$ are distributed in a disk of radius $R$. Each node $v$ thus has hyperbolic polar coordinates $(r_v, \theta_v)$ with $r_v \in [0, R]$ and $\theta_v \in [0, 2\pi]$. Let $\delta$ be the average node degree. And let $n$ and $-\alpha$ denote the number of nodes in $V$ and the negative curvature, respectively. All nodes are randomly distributed points with radial density $p(r) = \alpha \frac{\sinh(\alpha r)}{\cosh(\alpha R) - 1}$ and uniform by angle, where $R = 2 \log \frac{n}{\delta}$. It is well known that such uniform random graphs in the hyperbolic plane yield a power-law distribution for the node degrees (Aldecoa et al., 2015).

## 2.1 ALL-PAIRS NEAR-SHORTEST PATH PROBLEM (APNSP)

The objective of APNSP routing problem is to locally compute for all node pairs of any graph $G \in \mathbb{G}$ their near-shortest path. Here, near-shortest path is defined as one whose length is within a user-specified factor ($\geq 1$) of the shortest path length.

Formally, let $d_e(O, D)$ denote the distance between two endpoints $O$ and $D$, and $d_{sp}(O, D)$ denote the length of the shortest path between these endpoints. Further, let $\zeta(O, D)$ denote the path stretch of the endpoints, i.e., the ratio $\frac{d_{sp}(O,D)}{d_e(O,D)}$.

**The APNSP Problem.** Learn a routing policy $\pi$ such that, for any graph $G = (V, E) \in \mathbb{G}$ and any origin-destination pair $(O, D)$ where $O, D \in V$, $\pi(O, D, v) = u$ finds $v$'s next forwarder $u$ and in turn yields the routing path $p(O, D)$ with path length $d_p(O, D)$ that with high probability is a near-shortest path. In other words, $\pi$ optimizes the accuracy of $p(O, D)$ as follows:

$$\max \; Accuracy_{G,\pi} = \frac{\sum_{O,D \in V} \eta(O, D)}{|V|^2}, \tag{1}$$

$$s.t. \; \eta(O, D) = \begin{cases} 1, & if \; \frac{d_p(O,D)}{d_{sp}(O,D)} \leq \zeta(O, D)(1 + \epsilon) \\ 0, & otherwise \end{cases} \tag{2}$$

Note that the user-specified factor for APNSP is $\zeta(O, D)(1 + \epsilon)$, where $\epsilon \geq 0$.

## 2.2 MDP FORMULATION FOR THE APNSP PROBLEM

To solve APNSP, we first formulate it as a Markov decision process (MDP) problem that learns to choose actions that maximize the expected future reward. In general, an MDP consists of a set of states $S$, a set of actions $A(s)$ for any state $s \in S$, an instantaneous reward $r(s, a)$, indicating the immediate reward for taking action $a \in A(s)$ at state $s$, and a state transition function $P(s'|s, a)$ which characterizes the probability that the system transits to the next state $s' \in S$ after taking action $a$ at the current state $s \in S$. To simplify the routing behavior in the problem, the state transition is assumed to be deter-

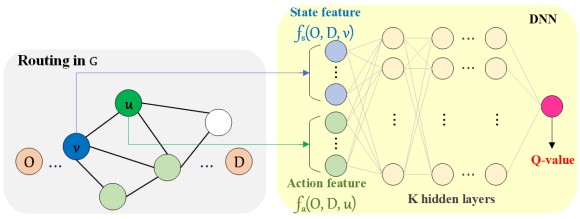

Figure 1: Schema for solution using DNN to predict $Q$-values for selecting the routing forwarder.

ministic. Specifically, each state $s$ represents the features of a node holding a packet associated with an origin-destination pair $(O, D)$, and an action $a \in A(s)$ indicates the routing behavior to forward the packet from the node to one of its neighbors. Given the current state $s$ and an action $a \in A(s)$ which selects one neighbor as the next forwarder, the next state $s'$ is determined as the features of the selected neighbor such that the probability $P(s'|s, a)$ is always one. The tuple $(s, a, r, s')$ is observed whenever a packet is forwarded.

In addition, we define the $Q$-value to specify the cumulative future reward from state $s$ and taking action $a$: $Q(s_t = s, a_t = a) = \sum_{i=t}^{L} \gamma^{i-t} r(s_i, a_i)$, where $\gamma, 0 \leq \gamma \leq 1$, is the discount factor. When

$\gamma = 0$, the instantaneous reward is considered exclusively, whereas the future rewards are treated as equally important as the instantaneous reward in the $Q$-value if $\gamma = 1$. In the APNSP problem, we define the instantaneous reward $r(s, a)$ as the negative length of the corresponding edge $(s, s')$, and set $\gamma = 1$. Therefore, the optimal $Q$-value, $Q^*(s, a)$, is equal to the cumulative negative length of the shortest path from $s$ to the destination.

For solving APNSP, we seek to learn the optimal $Q$-value through a data-driven approach with DNNs. As depicted in Figure 1, each state $s$ and action $a$ will be embedded into a set of input features denoted by $f_s(s)$ and $f_a(a)$, respectively. A DNN will be learned to approximate the optimal $Q$-value given the input features, based on which a near-shortest path routing policy can be obtained by choosing actions with largest $Q$-values.

## 2.3 DESIGN OF INPUT FEATURES

To learn a routing protocol that generalizes across graphs with different scales, densities, and topologies, the input features of the DNN should be designed to be independent of global network configurations, including the identity of nodes and of packets. Recall that each node knows its own coordinates and the coordinates of the origin and the destination.

Input features based on distance and on stretch factor have been found useful in local geographic routing protocols. Accordingly, we design the input features, including state and action features, as follows:

- State feature, $f_s(O, D, v)$. For a packet with its specified origin $O$ and destination $D$ at node $v$, the state features are the vectors with the elements below.

    1. **Distance to destination,** $dist(v, D)$**:** the distance between $v$ and $D$.

    2. **Stretch factor,** $sf(O, D, v) = \frac{dist(O,v)+dist(v,D)}{dist(O,D)}$**:** the stretch of the indirect distance between $O$ and $D$ that is via $v$ with respect to the direct distance between $O$ and $D$.

- Action feature, $f_a(O, D, a) = f_s(O, D, u)$. The feature for the action that forwards a packet from $v$ to $u \in nbr(v)$, $f_a(O, D, a)$ is chosen to be the same as the state feature of $u$, $f_s(O, D, u)$.

Henceforth, we consider learning with two different combinations of input features, one with only $dist(v, D), dist(u, D)$ and the other with both $dist(v, D), dist(u, D)$ and $sf(O, D, v), sf(O, D, u)$.

## 3 ASSURING GENERALIZABILITY OF ROUTING POLICIES

We begin with a sufficient condition for how a routing policy, $\pi$, learned from a seed graph $G^* = (V^*, E^*) \in \mathbb{G}$, where $\mathbb{G}$ is the set of all uniform random graphs, with select samples generated from a subset of nodes in $V^*$, can generalize over other (and potentially all) graphs $G \in \mathbb{G}$.

A basis for generalizability in this setting is the concept of a ranking metric for each node $v \in V$ with respect to each $u \in nbr(v)$, where $nbr(v)$ denotes the set of $v$'s one-hop neighbors. Let $f_s : V \to \mathbb{R}^I$ be a map of $v \in V$ to its state features (of cardinality $I$) and let $f_a : V \to \mathbb{R}^J$ be a map of $u \in nbr(v)$ to its action features (of cardinality $J$). We define a ranking metric $m(f_s(O, D, v), f_a(O, D, u)) \in \mathbb{R}$ to be a *linear* function over the input features associated with $v$ and $u$. For notational convenience, given an ordering $\langle u_0, ..., u_d \rangle$ of all nodes in $nbr(v)$, let $X_v = \{(f_s(O, D, v), f_a(O, D, u_0)), ..., (f_s(O, D, v), f_a(O, D, u_d))\}$, denote the set of input vectors for each corresponding neighbor $u_k \in nbr(v), 0 \le k \le d$. Also, let $Y_v = \{Q(v, u_0), ..., Q(v, u_d)\}$ denote the corresponding set of $Q$-values.

Consider then a sufficient condition on the relation between the local ranking metric $m$ and the corresponding $Q$-values set $Y_v$, namely, the global ranking metric, to learn a DNN model for ranking the neighbors $u \in nbr(v)$ according to their $Q$-values.

**Theorem 0** (Learnability). *Let $v$ be any node in $V$ for which ranking metric $m(f_s(O, D, v), f_a(O, D, u))$ satisfies the following property,* **RankPres***:*

>*If $\langle m(f_s(O, D, v), f_a(O, D, u_0)), ..., m(f_s(O, D, v), f_a(O, D, u_d)) \rangle$ is mono-tonically increasing[1], then $\langle Q(v, u_0), ..., Q(v, u_d) \rangle$ is monotonically increasing.*

*There exists a learnable DNN $H$, $H : \mathbb{R}^{I+J} \rightarrow \mathbb{R}$, with training samples $\langle X_v, Y_v \rangle$, that achieves optimal ranking of all $u \in nbr(v)$, i.e., its output for the corresponding neighbors of $v$, $\langle H(f_s(O, D, v), f_a(O, D, u_0)), ..., H(f_s(O, D, v), f_a(O, D, u_d)) \rangle$, is monotonically increasing.*

Next, we lift the sufficient condition to provide a general basis, first, for ranking the neighbors of all nodes in a graph according to their optimal shortest paths, from only the samples derived from one (or a few) of its nodes; and second, for similarly ranking the neighbors of all graphs in $\mathbb{G}$.

**Theorem 1** (Cross-Node Generalizability). *For any graph $G$, $G = (V, E)$, if there exists a ranking metric $m(f_s(O, D, v), f_a(O, D, u))$ that satisfies the RankPres property for all $v \in V$, then an optimal ranking policy for all $v \in V$ is learnable with only a subset of training samples $\langle X_{V'}, Y_{V'} \rangle$, where $V' \subseteq V$, $X_{V'} = \bigcup_{v \in V'} X_v$, and $Y_{V'} = \bigcup_{v \in V'} Y_v$.*

Note that if the $Q(v, u)$ value corresponds to the optimal (shortest) path $Q$-value for each $(v, u)$ pair, then the DNN indicated by Theorem 1 achieves an optimal routing policy for all nodes in $V$. Note also in this case that if the ranking metric $m$ satisfies *RankPres* not for all nodes but for almost all nodes, a policy learned from samples from one or more nodes $v$ that satisfy *RankPres* may not achieve optimal routing for all nodes. Nevertheless, if the relative measure of the number of nodes that do not satisfy *RankPres* to the number of nodes that do satisfy *RankPres* is small, then with high probability the policy achieves near-optimal routing.

**Theorem 2** (Cross-Graph Generalizability). *If there exists a ranking metric $m(f_s(O, D, v), f_a(O, D, u))$ that satisfies the RankPres property for the nodes in all graphs $G \in \mathbb{G}$, then an optimal ranking policy is learnable by using training samples from one or more nodes in one or more chosen seed graph(s) $G^* \in \mathbb{G}$.*

Again, if Theorem 2 is considered in the context of $Q$-values corresponding to optimal shortest paths, the learned routing policy $\pi$ generalizes to achieving optimal routing over all graphs $G \in \mathbb{G}$. And if we relax the requirement that *RankPres* holds for all nodes of all graphs in $\mathbb{G}$ to only requiring that for almost all graphs $G \in \mathbb{G}$, there is a high similarity between the ranking metric $m$ and the optimal $Q$-value, then with high probability the policy achieves near-optimal routing.

Proofs of the above-mentioned theorems are relegated to Appendix A.

**Proposition 1.** *For APNSP, there exists a local ranking metric $m_1(f_s(O, D, v), f_a(O, D, u))$ of the form $w_1.dist(v, D) + w_2.dist(u, D)$ based on the **distance** input feature that satisfies the RankPres property for almost all nodes in almost all graphs $G$.*

*Also, there exists a local ranking metric $m_2(f_s(O, D, v), f_a(O, D, u))$ of the form $w_1.dist(v, D) + w_2.sf(O, D, v) + w_3.dist(u, D) + w_4.sf(O, D, u)$ based on both **distance** and **stretch factor** input features that satisfies the RankPres property for almost all nodes in almost all graphs $G$.*

We empirically validate Proposition 1, as presented in Appendix B. RankPres is quantified in terms of Ranking Similarity; high ranking similarity implies that RankPres holds with high probability. We show that Proposition 1 holds for both Euclidean and hyperbolic spaces with respectively chosen weights $w$ for $m_1$ and $m_2$.

It follows that an efficient generalizable policy for APNSP is feasible for each of the two chosen input feature sets, given the existence of respective ranking metrics that with high probability satisfy RankPres. For APNSP, the optimal $Q(v, u)$ values can be retrieved by calculating the length of the shortest path starting from $v$ toward $u$ until reaching the destination. A near-optimal routing policy may then be learned via supervised learning on a single seed graph.

---

[1]By monotonic increasing order in $\langle m(f_s(O, D, v), f_a(O, D, u_0)), ..., m(f_s(O, D, v), f_a(O, D, u_d)) \rangle$, we mean $m(f_s(O, D, v), f_a(O, D, u_0)) \leq ... \leq m(f_s(O, D, v), f_a(O, D, u_d))$.

# 4 SINGLE GRAPH LEARNING ALGORITHM

## 4.1 SELECTION OF SEED GRAPH AND GRAPH SUBSAMPLES

To achieve both cross-graph generalizability and cross-node generalizability, we develop a knowledge-guided mechanism with the following two selection components:

**Seed Graph Selection.** The choice of seed graph depends primarily on the analysis of cross-node generalizability (Theorem 1) across a sufficient set of uniform random graphs with diverse sizes and densities/average node degrees. In Figures 10 and 11 in Appendix B.2, we empirically show that, with the use of distance and stretch factor, a good seed graph is likely to exist in a set of graphs with small size (e.g., 50) and high density (e.g., 5) in the Euclidean space and high average node degree (e.g., 4) in the hyperbolic space.

There may be applications where analysis of large (or full) graphs is not always possible. In such situations, given a graph $G$, an alternative choice of seed graph can be from a small subgraph $G' = (V', E'), V' \subset V, E' \subset E$ with relatively high cross-node generalizability. Note that, in Theorem 1, for a graph $G = (v, E)$ satisfying the *RankPres* property for all $v \in V$, *RankPres* still holds for $v' \in V'$ in a subgraph $G' = (V', E')$ of $G$. This is because the learnable function $m$ still preserves the optimal routing policy for all nodes in a subset of $nbr(v)$.

**Graph Subsamples Selection.** We provide the following scheme of subsample selection for a given graph $G = (V, E)$ to choose a set of $\phi$ nodes for generating $\phi\delta$ training samples.

1. Select an origin and destination pair $(O, D), O, D \in V$.
2. Select $\phi$ nodes, $v_0, ..., v_{\phi-1} \in V \setminus D$.
3. For each chosen node $v_0, ..., v_{\phi-1}$, respectively, collect the subsamples $\langle X, Y \rangle$, where
   $X = \bigcup_{v \in \{v_0, ..., v_{\phi-1}\} \wedge u \in nbr(v)} \{\langle f_s(O, D, v), f_a(O, D, u) \rangle\}$ and
   $Y = \bigcup_{v \in \{v_0, ..., v_{\phi-1}\} \wedge u \in nbr(v)} \{Q^*(v, u)\}$.

In Appendix F, we analyze the complexity of graph subsampling. An alternative for seed nodes search is also provided to limit the search complexity.

## 4.2 SUPERVISED LEARNING FOR APNSP WITH OPTIMAL Q-VALUES

Given the dataset $\langle X, Y \rangle$ collected from a seed graph, we train a DNN based on supervised learning to capture the optimal ranking policy. Specifically, suppose the DNN $H$ is parameterized by $\Theta$. We seek to minimize the following loss function:

$$\min_{\Theta} \sum_{\langle X, Y \rangle} \|H_{\Theta}(f_s(O, D, v), f_a(O, D, u)) - Q^*(v, u)\|^2. \tag{3}$$

Note that we assume that the optimal $Q$-values are known for the seed graph in the supervised learning above, which can be obtained based on the shortest path routing policies of the seed graph. By leveraging these optimal $Q$-values and supervised learning on the seed graph, a generalized routing policy is learned for APNSP routing over almost all uniform random graphs in both Euclidean and hyperbolic spaces, as we validate in the experiments in Section 5.

## 4.3 REINFORCEMENT LEARNING FOR APNSP

For the case where the optimal $Q$-values of graphs are unknown, we solve the APNSP problem using RL. Using the same input features and seed graph selection procedure, the RL algorithm continuously improves the quality of $Q$-value estimations by interacting with the seed graph.

In contrast to the supervised learning algorithm, where we collect only a single copy of data samples from a set of chosen (shortest path) nodes once before training, in RL new training data samples from nodes in a shortest path, predicted by most recent training episode (i.e., based on the current $Q$-value estimation), are collected at the beginning of each training episode. Remarkably, the generalizability of the resulting RL routing policy across almost all uniform random graphs in Euclidean and hyperbolic spaces is preserved. The details of the RL algorithm, named as RL-APNSP-ALGO, are shown in Algorithm 1 in Appendix C.

# 5 ROUTING POLICY PERFORMANCE FOR SCALABILITY AND ZERO-SHOT GENERALIZATION

In this section, we discuss implementation of our machine learned routing policies and evaluate their performance in predicting all-pair near-shortest paths for graphs across different sizes and densities over Euclidean and hyperbolic spaces in Python3. We use PyTorch 2.0.1 (The Linux Foundation, 2023) on the CUDA 11.8 compute platform to implement DNNs as shown in Figure 1. Table 3 in Appendix D.1 shows our simulation parameters for training and testing the routing policies.

## 5.1 COMPARATIVE EVALUATION OF ROUTING POLICIES

We compare the performance of the different versions of *Greedy Tensile* policies obtained using the following approaches:

**Supervised ($\phi$=3) / Supervised (all)**: Supervised learning from appropriately chosen seed graph $G^*$ using graph subsamples selection from $\phi$=3 or all nodes.

**RL ($\phi$=3) / RL (all)**: RL from appropriately chosen seed graph $G^*$ using graph subsamples selection from $\phi$=3 or all nodes.

**GF**: Greedy forwarding that forwards packets to the one-hop neighbor with the minimum distance to the destination.

Note that for a given set of input features, both supervised learning and reinforcement learning schemes use the same DNN configuration to learn the routing policies. By using the subsampling mechanism, not only the sample complexity but also the training time will be significantly reduced in Supervised ($\phi$=3) and RL ($\phi$=3) compared to those in Supervised (all) and RL (all), respectively.

## 5.2 ZERO-SHOT GENERALIZATION OVER DIVERSE GRAPHS

To evaluate the scalability and generalizability of the policies, we directly (i.e., without any adaptation) test the policies learned from the seed graph $G^*$ on new uniform random graphs with different combinations of $(N_{test}, \rho_{test})$ in Euclidean spaces and $(N_{test}, \delta_{test})$ in hyperbolic spaces. We select 20 random graphs for each pair and calculate the average prediction accuracy over these $20N_{test}^2$ shortest paths.

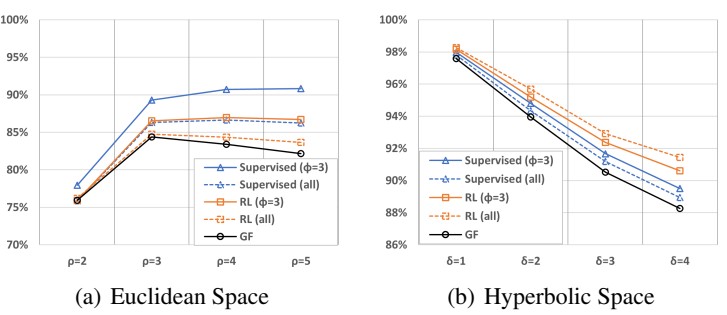

(a) Euclidean Space          (b) Hyperbolic Space

Figure 2: Average APNSP prediction accuracy across graph sizes with various $\rho$ and $\delta$ for *Greedy Tensile* policies.

For the DNNs with input $dist(v, D)$ and $dist(u, D)$, the tests confirm that the performance of all the learned policies exactly match the prediction accuracy of Greedy Forwarding in both Euclidean and hyperbolic spaces. (In Appendix E, we symbolically interpret the learned DNN with input $dist(v, D)$ and $dist(u, D)$ to show it can be reduced to GF.)

For the *Greedy Tensile* DNNs with input $\langle dist(v, D), sf(O, D, v), dist(u, D), sf(O, D, u) \rangle$, we plot in Figure 2 the respective average prediction accuracies across graph sizes in $\{27, 64, 125, 216\}$ with density in $\{2, 3, 4, 5\}$ in Euclidean space and average node degree in $\{1, 2, 3, 4\}$ in hyperbolic space.

In Euclidean space, the Supervised ($\phi$=3) approach achieves the best performance among all the approaches. In particular, compared to GF, the Supervised ($\phi$=3) policy improves the accuracy up to 9% over GF, whereas the other learned policies show at least comparable performance in low density

graphs ($\rho = 2$) and achieve an improvement of up to 5% in graphs with $\rho \geq 3$. The performance gap between the DNNs and GF increases as the network density increases to a high level (e.g., $\rho = 5$), wherein GF was believed to work close to the optimal routing.

In hyperbolic space, the RL (all) policy improves the accuracy up to 3% over GF, whereas the other learned policies show at least comparable performance in low degree graphs ($\delta = 1$) and achieve an improvement of up to 2% in graphs with $\delta \geq 2$. To the best of our knowledge, we are the first to provide routing policies that outperform GF in almost all random graphs; recall GF was shown to find almost optimal shortest path in scale-free topologies (Papadopoulos et al., 2010).

## 6 SYMBOLIC INTERPRETABILITY OF LEARNED MODEL

In this section, we symbolically interpret the learned model for *Greedy Tensile* routing, achieving a two orders of magnitude reduction in its operational complexity. Figure 3 plots the output of its DNN, towards explaining the learned policy[2]. Since $dist(v, D)$ and $sf(O, D, v)$ stay unchanged for a fixed routing node $v$ at a given time, we plot the shape of the ranking metric (z-axis) of the learned DNN according to varying $sf(O, D, v)$ (x-axis) and $dist(u, D)$ (y-axis) in the figure.

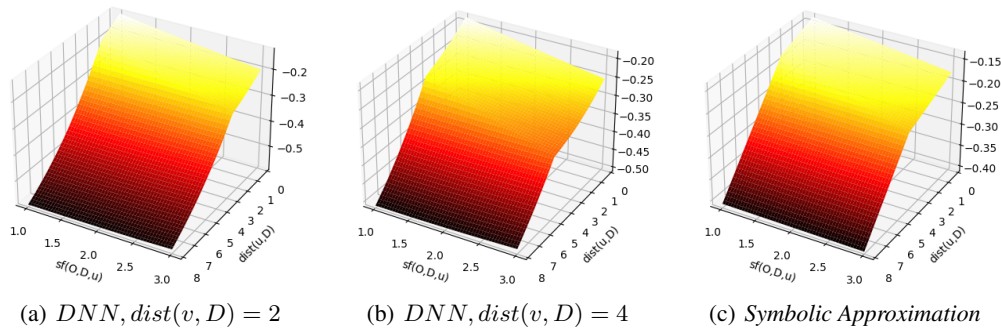

$$(a)\ DNN, dist(v, D) = 2 \qquad (b)\ DNN, dist(v, D) = 4 \qquad (c)\ Symbolic\ Approximation$$

Figure 3: The shape of ranking metrics of the *Greedy Tensile* DNN and its two-linear action Symbolic Approximation policy, given $sf(O, D, v) = 1.2$. The x and y axes represent $sf(O, D, u)$ and $dist(u, D)$, and the z axis is the ranking metric for routing.

Figures 3(a) and 3(b) show that the *Greedy Tensile* DNN has two planes separated by a transition boundary that varies as $dist(v, D)$ changes. The two planes can be respectively approximated by two different linear functions. The first function (for the upper plane) prefers both smaller $sf(O, D, u)$ and $dist(u, D)$. The second (for the lower plane) significantly prioritizes smaller $dist(u, D)$. We find that the two functions that approximate the *Greedy Tensile* DNN can be symbolically represented by a guarded command:

$$z = \begin{cases} -0.013 dist(v, D) - 0.023 sf(O, D, u) - 0.012 dist(u, D) - 0.063, \\ \quad if\ dist(u, D) < 1.020 dist(v, D) + 0.567 sf(O, D, u) - 0.690 \\ 0.025 dist(v, D) - 0.002 sf(O, D, u) - 0.044 dist(u, D) - 0.146,\ \ otherwise. \end{cases} \quad (4)$$

The weights of the guarded command are calculated using linear regression. Its first action assigns dominant weights to both $sf(O, D, u)$ and $dist(u, D)$, while its second action gives almost negligible weight to $sf(O, D, u)$ compared to the weight of $dist(uD)$. The shape of ranking metrics of the guarded command given $dist(v, D) = 4$ is shown in Figure 3(c), which has a two-plane surface similar to Figure 3(b). Figure 14 in Appendix D.3 shows that the accuracy of the two-linear action policy using Equation 4 is close to that of *Greedy Tensile* DNN.

The simplified two-linear-action policy also has substantially reduced operation complexity. Whereas the *Greedy Tensile* DNN requires at least $\Omega \times N_e[1] \times N_e[2] (= 4 * 200 * 4)$ multiplications to output the $Q$-value for a given (state, action) pair, where $\Omega$ represents the number of

---

[2]Since *Greedy Tensile* models in Euclidean and hyperbolic spaces have a similar shape, we only visualize the learned model in the Euclidean space here.

input features of the DNN and $N_e[i]$ denotes the number of neurons in the $i$-th hidden layer, the two-linear-action policy needs less than ten multiplications.

## 7 RELATED WORK

**Feature Selection for Routing**. A classic feature for local routing comes from greedy forwarding (Finn, 1987), where the distance to the destination node (in an Euclidean or hyperbolic metric space) is used to optimize forwarder selection. It has been proven that this feature achieves nearly optimal routing in diverse network configurations, including scale-free networks (Kleinberg, 2000; Papadopoulos et al., 2010). A stretch bound on routing paths using greedy forwarding is investigated in diverse models with or without the assumption of unit disk graphs (Flury et al., 2009; Tan et al., 2009b; Tan & Kermarrec, 2011; Tan et al., 2009a; Won & Stoleru, 2014). Other features for forwarder selection include Most Forward within Radius (MFR) (Takagi & Kleinrock, 1984), Nearest with Forwarding Progress (NFP) (Hou & Li, 1986), the minimum angle between neighbor and destination (aka Compass Routing) (Kranakis, 1999), and Random Progress Forwarding (RPF) (Nelson & Kleinrock, 1984).

Network domain knowledge has also been used to guide search efficiency in routing protocols. A recent study (Chen et al., 2023) shows that searching for shortest paths in uniform random graphs can be restricted to an elliptic search region with high probability. Its geographic routing protocol, *QF-Geo*, uses node Stretch Factor as an input feature to determine whether a node's neighbors lie in the search region and to forward packets only within the predicted elliptic region.

**Generalizability of Machine Learned Routing**. Only recently has machine learning research started to address generalizability in routing contexts. For instance, generalizability to multiple graph layout distributions, using knowledge distillation, has been studied for a capacitated vehicle routing problem (Bi et al., 2022). Some explorations have considered local search: i.e., wireless network routing strategies via local search based on deep reinforcement learning (Manfredi et al., 2021; 2022) have been shown to generalize to other networks of up to 100 nodes, in the presence of diverse dynamics including node mobility, traffic pattern, congestion, and network connectivity. Deep learning has also been leveraged for selecting an edge set for a well-known heuristic, Lin-Kernighan-Helsgaun (LKH), to solve the Traveling Salesman Problem (TSP) (Xin et al., 2021). The learned model generalizes well for larger (albeit still modest) sized graphs and is useful for other network problems, including routing. Likewise, graph neural networks and learning for guided local search to select relevant edges have been shown to yield improved solutions to the TSP (Hudson et al., 2021). In related work, deep reinforcement learning has been used to iteratively guide the selection of the next solution for routing problems based on neighborhood search (Wu et al., 2021).

## 8 CONCLUSIONS AND FUTURE WORK

We have shown that guiding machine learning with domain knowledge can lead to the rediscovery of a well-known routing policy (somewhat surprisingly), in addition to discovering a new routing policy, that perform well in terms of complexity, scalability, and generalizability. The theory we have presented in the paper is readily extended to other classes of graphs (such as non uniform cluster distributions), ranking metrics that are nonlinear, and MDP actions that span multiple neighbors. Thus, albeit our illustration intentionally uses relatively familiar input features and local routing architectures, richer domain theory will be useful to guide machine learning of novel routing algorithms. Moreover, the routing policies are likely to be competitive for richer classes of graphs than the class of uniform random graphs on which we have focused our validation.

While samples from nodes of a single seed graph suffice for generalizable learning, in practice, learning from multiple seed graphs may be of interest. For instance, if an ideal seed graph is not known a priori, online learning from better or multiple candidate seed graphs as they are encountered may be of interest for some applications. Along these lines, we recall that the set of ideal (and near ideal) seed graphs is relatively large in the problem we considered. One way to relax the knowledge of ideal seed graphs is to leverage online meta-learning, for learning a good model initialization and continuing to improve the initialization based on better seed graphs as they are encountered. Towards this end, we have also been studying the merits of efficiently fine tuning the model for the target graph as an alternative to zero-shot generalization.

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

## A    PROOF OF THEOREM

In this section, we provide the proof of the three theorems stated in Section 3.

1. Proof of Theorem 0.

   *Proof.* Since the ranking metric $m$ is a linear function, a single neuron DNN $H$ can achieve the optimal ranking by weighing each input feature with the weight corresponding to that feature in $m$. □

2. Proof of Theorem 1.

   *Proof.* From Theorem 0, $m$ can be learned by a DNN with samples generated from any $v \in V$ (or, more generally, from any set of nodes $V' \subseteq V$). Since the RankPres property holds for all $v \in V$, the resulting DNN achieves an optimal ranking of all nodes in $V$ and hence for $G$. □

3. Proof of Theorem 2.

   *Proof.* From Theorem 2, learning from $G^*$ with training samples chosen from one or more nodes in $G^*$ achieves an optimal ranking policy $\pi$ for all nodes in $G^*$. The soundness of the subsampling argument suffices for the same policy to achieve optimal ranking for nodes in any graph $G \in \mathbb{G}$, as long as there exists a common ranking metric $m$ with the RankPres property holding for all graphs in $\mathbb{G}$. □

## B    RANKING SIMILARITY BETWEEN LOCAL AND GLOBAL METRICS

Based on the theory of Section 3, we now investigate the existence of a ranking metric $m$ for the first input feature, as well as the pair of input features, and demonstrate that in both cases the ranking similarity is close to 1 across nodes in almost all graphs $G$, regardless of their size and density. The results thus empirically corroborate the feasibility of routing policy generalization from single graph learning, and also guide the selection of seed graphs and training samples.

Let $SIM_v(m, Q^*) \in [0, 1]$ denote the *ranking similarity* between the ranking metric $m$ and the optimal $Q$-values ($Q^*$) at node $v$. Moreover, let $SIM_G(m, Q^*) \in [0, 1]$ denote the average ranking similarity between $m$ and $Q^*$ across all nodes $v \in V$ in a given graph $G = (V, E)$, i.e., $SIM_G(m, Q^*) = \frac{\sum_{v \in V} SIM_v(m, Q^*)}{|V|}$.

Conceptually, $SIM_v(m, Q^*)$ should tend to 1 as the order of neighbors in the sorted ascending order of $m(f_s(O, D, v), f_a(O, D, u)), u \in nbr(v)$, comes closer to matching the order of the neighbors in the sorted ascending order of $Q^*(v, u)$. More specifically, we adopt Discounted Cumulative Gain (DCG) (Järvelin & Kekäläinen, 2002) to formally define the ranking similarity. The idea of DCG is to evaluate the similarity between two ranking sequences by calculating the sum of graded relevance of all elements in a sequence. First, a sorted sequence of $Q^*(v, u)$ with length $L = |nbr(v)|$, $u \in nbr(v)$, is constructed as the ideal ranking, denoted by $A$. For each position $i$ in $A$, we assign a graded relevance $rel_A[i]$ by following the rule: $rel_A[i] = (L - i)^2, i = 0...L - 1$[3]. The value of DCG accumulated at a particular rank position $\tau$ is defined as:

$$DCG_\tau = \sum_{i=1}^{\tau} \frac{rel[i]}{\log_2(i+1)}.$$

For example, let $A = [4, 1, 3, 2, 5]$. The corresponding $rel_A[i]$ and $\frac{rel_A[i]}{\log_2(i+1)}$ values are shown in Table 1. Then $A$'s $DCG_3 = 25 + 10.095 + 4.5 = 39.595$.

Next, a sorted sequence of $m(f_s(O, D, v), f_a(O, D, u))$ with length $L = |nbr(v)|$, $u \in nbr(v)$, is constructed as the estimated ranking, denoted by $B$. Let $B = [1, 2, 4, 5, 6]$. The graded relevance

---

[3]The assignment of graded relevance could be any way to decrease the value from left to right positions. Here we use squared value to assign dominant weights to the positions close to leftmost.

Table 1: An example of DCG calculation for an ideal ranking $A = [4, 1, 3, 2, 5]$.

| $i$ | $A[i]$ | $rel_A[i]$ | $\log_2(i+1)$ | $\frac{rel_A[i]}{\log_2(i+1)}$ |
|---|---|---|---|---|
| 1 | 4 | 25 | 1 | 25 |
| 2 | 1 | 16 | 1.585 | 10.095 |
| 3 | 3 | 9 | 2 | 4.5 |
| 4 | 2 | 4 | 2.322 | 1.723 |
| 5 | 5 | 1 | 2.807 | 0.387 |

Table 2: An example of DCG calculation for an estimated ranking $B = [1, 2, 4, 5, 6]$.

| $j$ | $B[j]$ | $rel_B[j]$ | $\log_2(j+1)$ | $\frac{rel_B[j]}{\log_2(j+1)}$ |
|---|---|---|---|---|
| 1 | 1 | 16 | 1 | 16 |
| 2 | 2 | 4 | 1.585 | 2.524 |
| 3 | 4 | 25 | 2 | 12.5 |
| 4 | 5 | 1 | 2.322 | 0.431 |
| 5 | 6 | 0 | 2.807 | 0 |

for $B[j]$ depends on the position of $B[j]$ in $A$ and follows the rule:

$$rel_B[j] = \begin{cases} rel_A[i], & if \ (\exists i \ : \ A[i] = B[j]) \\ 0, & otherwise \end{cases}$$

Then $B$'s corresponding $rel_B[j]$ and $\frac{rel_B[j]}{\log_2(j+1)}$ values are shown in Table 2. Accordingly, $B$'s $DCG_3 = 16 + 2.524 + 12.5 = 31.024$. The ranking similarity between $B$ and $A$ is calculated by the ratio of $B$'s DCG to $A$'s DCG, i.e., $\frac{31.024}{39.595} = 0.784$.

### B.1 RANKING SIMILARITY WITH DISTANCE INPUT FEATURE.

For $\langle f_s(O, D, v), f_a(O, D, u) \rangle = \langle dist(v, D), dist(u, D) \rangle$, we examine if there exists a linear function $m$ that yields high $SIM_v(m, Q^*)$ and $SIM_G(m, Q^*)$ across different network configurations. Let the ranking metric $m$ be $m(f_s(O, D, v), f_a(O, D, u)) = -dist(u, D)$. (Recall that $Q^*(f_s(O, D, v), f_a(O, D, u))$ is the cumulative negative length of the shortest path.)

In Figure 4, we plot $SIM_v(m, Q^*)$ for all $v, D \in V$ for a given random Euclidean graph with size in $\{50, 100\}$ and density in $\{3, 5\}$. In Figure 5, we plot $SIM_v(m, Q^*)$ for all $v, D \in V$ for a given random hyperbolic graph with size in $\{50, 100\}$ and average node degree in $\{2, 4\}$. Each point represents the value of $SIM_v(m, Q^*)$ for a given $v$ and $D$. All $|V|^2$ points are shown in ascending order. The sub-figures in Figures 4 and 5 illustrate that at least 75% of the points have high similarity ($\geq 90\%$) between $m$ and $Q^*$. According to Theorem 1, the distribution of $SIM_v(m, Q^*)$ implies that training samples collected from one (or a few) nodes in this large set can be sufficient to learn a near-optimal routing policy that achieves high accuracy across nodes. On the other hand, using training samples generated from the nodes with relatively low $SIM_v(m, Q^*)$ should be avoided. This observation motivates a knowledge-guide mechanism to carefully choose the data samples used for training.

In Figures 6 and 7, we show the distribution of $SIM_G(m, Q^*)$ in ascending order for a set of 100 graphs, respectively with size in $\{50, 100\}$ with density in $\{3, 5\}$ in the Euclidean space and average node degree in $\{2, 5\}$ in the hyperbolic space. Note that in high density Euclidean graphs (say density 5) and high degree hyperbolic graphs (say degree 4), all 100 graphs have high similarity ($\geq 90\%$) between $m$ and $Q^*$, implying that training with samples from almost any high density Euclidean graph and high degree hyperbolic graph can be sufficient to learn a routing policy with high performance. On the other hand, in low density Euclidean graphs (say density 3) and low degree hyperbolic graphs (say degree 2), there exist a small set of graphs with slightly low $SIM_G(m, Q^*)$, pointing to the importance of careful selection of seed graph(s) for training.

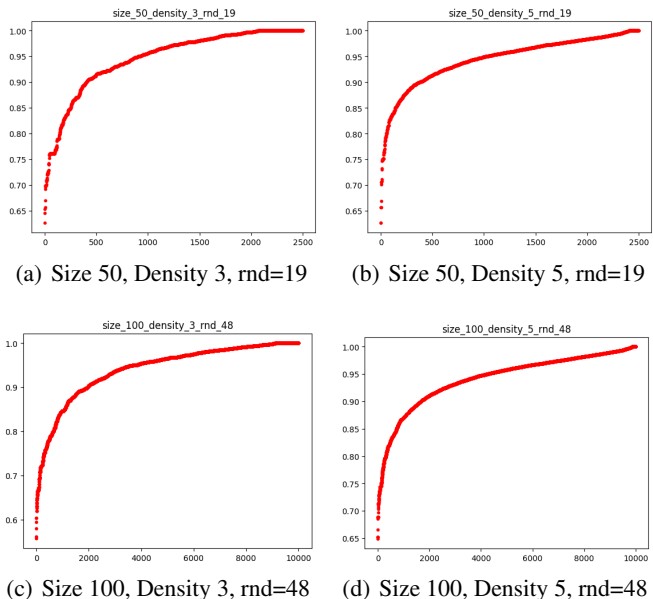

Figure 4: Distribution of $SIM_v(m, Q^*)$ given a uniform random graph in the Euclidean space with random seed (rnd), where $m$ is the ranking metric for distance $dist(u, D)$.

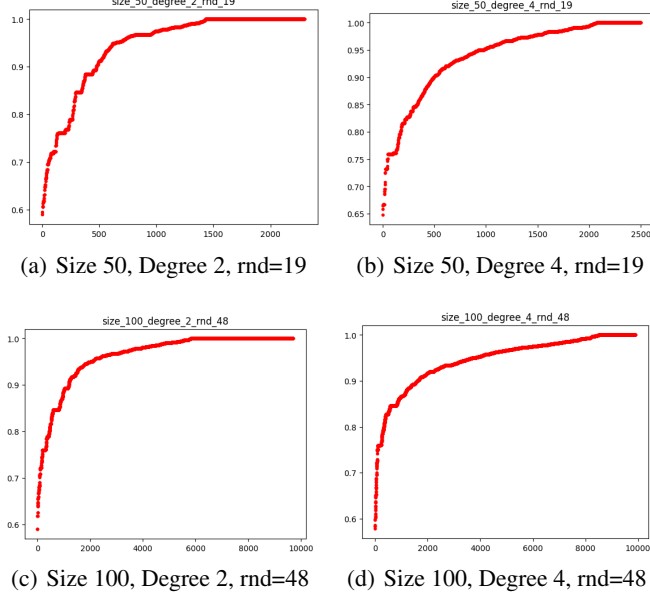

Figure 5: Distribution of $SIM_v(m, Q^*)$ given a uniform random graph in the hyperbolic space with random seed (rnd), where $m$ is the ranking metric for distance $dist(u, D)$.

In addition, we observe that the upper bound of $SIM_G(m, Q^*)$ decreases as the network size increases. These results implicitly show that, with respect to the chosen input feature, graphs with smaller size but higher density can be better seeds for learning generalized routing policies.

According to the results in Figures 4 and 5 and Figures 6 and 7, using distance implies the existence of a ranking metric that should with high probability satisfy cross-node generalizability and cross-graph generalizability. The ranking function, $m(f_s(O, D, v), f_a(O, D, u)) = -dist(u, D)$, or an

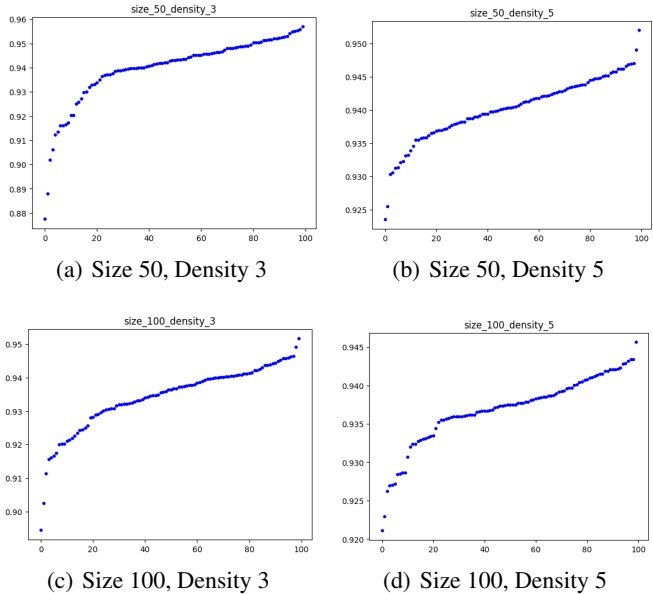

Figure 6: Distribution of $SIM_G(m, Q^*)$ across 100 uniform random graphs in the Euclidean space, where $m$ is the ranking metric for distance $dist(u, D)$.

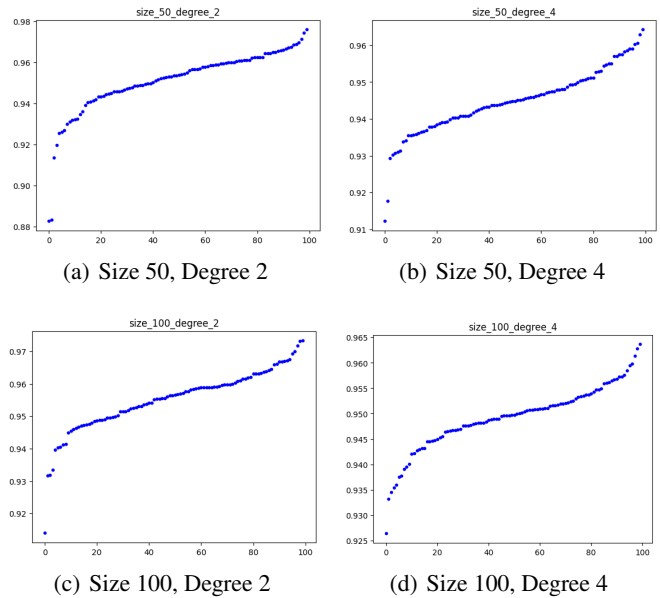

Figure 7: Distribution of $SIM_G(m, Q^*)$ across 100 uniform random graphs in the hyperbolic space, where $m$ is the ranking metric for Euclidean distance $dist(u, D)$.

analogue can be easily learned by a DNN, and then the learned routing policy should achieve high performance across uniform random graphs.

## B.2 RANKING SIMILARITY WITH DISTANCE AND STRETCH FACTOR INPUT FEATURES.

We examine if there exists a linear function $m$ that yields high $SIM_v(m, Q^*)$ and $SIM_G(m, Q^*)$ across different network configurations, for $\langle f_s(O, D, v), f_a(O, D, u) \rangle = \langle dist(v, D), sf(O, D, v), dist(u, D), sf(O, D, u) \rangle$. Let the ranking metric $m$ as

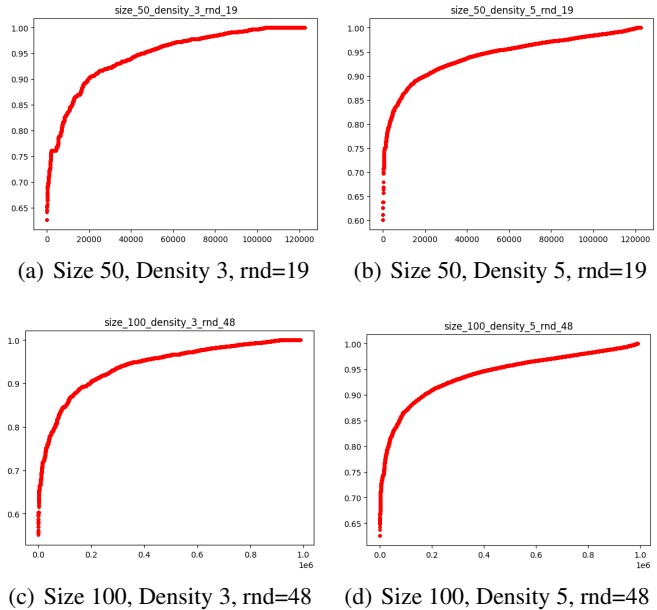

(a) Size 50, Density 3, rnd=19          (b) Size 50, Density 5, rnd=19

(c) Size 100, Density 3, rnd=48     (d) Size 100, Density 5, rnd=48

Figure 8: Distribution of $SIM_v(m, Q^*)$ given a uniform random graph in the Euclidean space with a random seed (rnd), where $m$ is the ranking metric for Euclidean distance $dist(u, D)$ and stretch factor $sf(O, D, u)$.

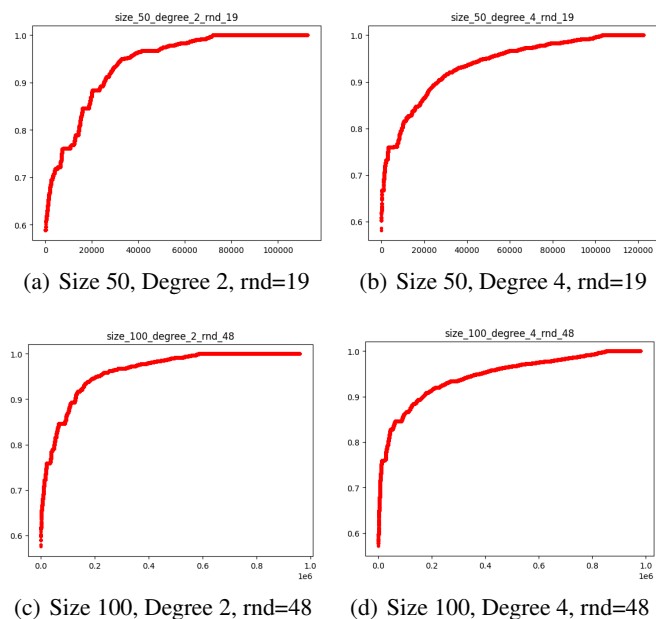

(a) Size 50, Degree 2, rnd=19          (b) Size 50, Degree 4, rnd=19

(c) Size 100, Degree 2, rnd=48     (d) Size 100, Degree 4, rnd=48

Figure 9: Distribution of $SIM_v(m, Q^*)$ given a uniform random graph in the hyperbolic space with a random seed (rnd), where $m$ is the ranking metric for Euclidean distance $dist(u, D)$ and stretch factor $sf(O, D, u)$.

$m(f_s(O, D, v), f_a(O, D, u)) = -0.875dist(u, D) - 0.277sf(O, D, u)$ in the Euclidean graphs and let $m$ be $m(f_s(O, D, v), f_a(O, D, u)) = -dist(u, D) - sf(O, D, u)$ in the hyperbolic graphs. Note that $f_s(O, D, v)$ does not affect the sorting of $m(f_s(O, D, v), f_a(O, D, u))$. For convenience, we assign zero weight for $dist(v, D)$ and $sf(O, D, v)$ in $m$.

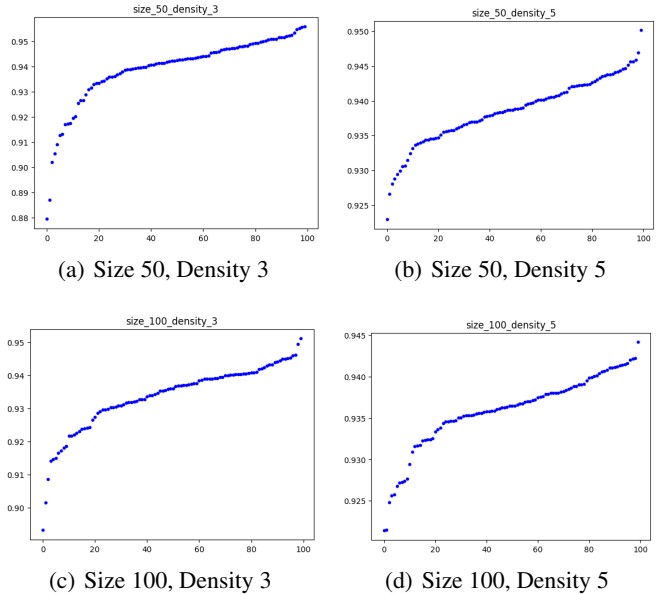

(a) Size 50, Density 3      (b) Size 50, Density 5

(c) Size 100, Density 3      (d) Size 100, Density 5

Figure 10: Distribution of $SIM_G(m, Q^*)$ across 100 uniform random graphs in the Euclidean space, where $m$ is the ranking metric for distance $dist(u, D)$ and stretch factor $sf(O, D, u)$.

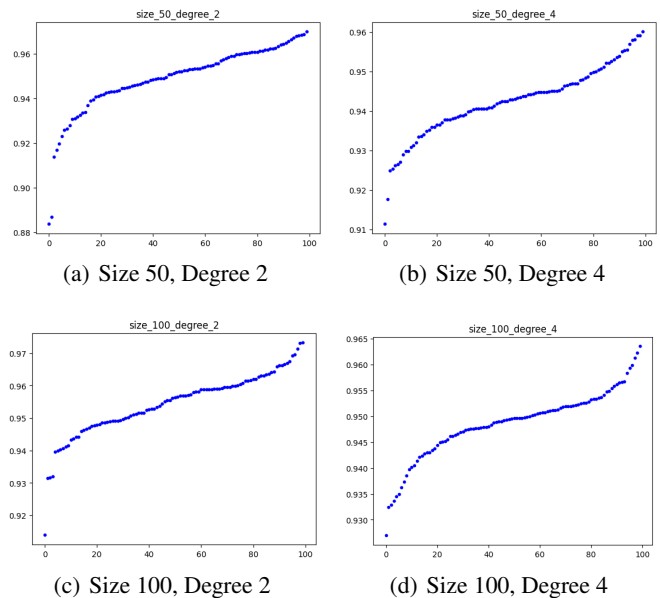

(a) Size 50, Degree 2      (b) Size 50, Degree 4

(c) Size 100, Degree 2      (d) Size 100, Degree 4

Figure 11: Distribution of $SIM_G(m, Q^*)$ across 100 uniform random graphs in the hyperbolic space, where $m$ is the ranking metric for distance $dist(u, D)$ and stretch factor $sf(O, D, u)$.

In Figures 8 and 10, we apply the same network configurations in the Euclidean graphs as in Figures 4 and 6 to plot the distribution of $SIM_v(m, Q^*)$ and $SIM_G(m, Q^*)$ for the proposed choice of $m$. In Figures 9 and 11, we apply the same network configurations in the hyperbolic graphs as in Figures 5 and 7. Note that because the stretch factor relies on $v, O, D \in V$, there are $|V|^3$ points plotted in Figureg 8 and 9 with ascending order. The sub-figures in Figures 8 and 9 demonstrate that at least 80% of the points have similarity above 90%, which is higher than the corresponding percentage of points in Figures 4 and 5. These observations indicate that using both distance and stretch factor assures the existence of ranking metric, and in turn implies learnability of a DNN that

with high probability achieves even better cross-node generalizability, compared to the one using only Euclidean distance in the input features.

Figures 10 and 11 shows a similar distribution of $SIM_G(m, Q^*)$ to Figures 6 and 7, respectively. In order to achieve cross-graph generalizability, we need to carefully select a seed graph for training, especially from graphs with moderate size and high density. Also, instead of using all the nodes to generate data samples, using a subset of nodes that avoid relatively low $SIM_v(m, Q^*)$ further improves the cross-node generalizability.

## C  ALGORITHM OF THE REINFORCEMENT LEARNING FOR APNSP

---
**Algorithm 1:** RL-APNSP-ALGO

---
1  Input: $nn$: randomly initialized DNN; $G^*$: seed graph; $\Phi$: set of chosen nodes; $O, D$: chosen origin and destination
2  **for** $episode = 1...EpiNum$ **do**
3      $X$:=[], $Y$:=[], $i := 0$;
4      **for** $v \in \Phi$ **do**
5          **for** $u \in nbr(v)$ **do**
6              $X[i] := \langle f_s(O, D, v), f_a(O, D, u)\rangle$;
7              Estimate $Q(v, u)$ using Equation 5;
8              $Y[i] := Q(v, u)$;
9              $i := i + 1$;
10          **end**
11      **end**
12      **for** $iter = 1...IterNum$ **do**
13          Train $nn$ with $\langle X, Y \rangle$ based on Equation 6;
14      **end**
15  **end**
16  **return** $nn$;

---

More specifically, in Algorithm 1, Lines 2 to 15 outline the sample selection and training procedure for each episode. The for-loop in Lines 4 to 11 generates the training data samples from set of chosen nodes, where the data labels $Y$, i.e., target $Q$-values that we train the DNN to fit for improving the estimation accuracy of optimal $Q$-values, are given by:

$$Q^{target}(s, a) = r(s, a) + \gamma \max_{a'} Q(s', a'). \tag{5}$$

Next, based on the collected dataset, in Lines 12 to 14, the DNN is trained for a fixed number of iterations to minimize the following loss function:

$$\min_{\Theta} \sum_{\langle X, Y\rangle} \|H_\Theta(f_s(O, D, v), f_a(O, D, u)) - Q^{target}(v, u)\|^2. \tag{6}$$

Since the target $Q$-values approach the optimal $Q$-values as the number of training episodes increases, minimizing Equation 6 will eventually lead to a learned model that nearly matches the supervised learning in Equation 3.

## D  DETAILED ROUTING POLICY PERFORMANCE EVALUATION

### D.1  DNN CONFIGURATION

Table 3 shows the simulation parameters for training and testing the routing policies in Euclidean and hyperbolic spaces.

### D.2  PERFORMANCE EVALUATION FOR *Greedy Tensile* POLICIES

Figure 12 shows the respective prediction accuracies of *Greedy Tensile* policies across graphs in Euclidean space whose size is in $\{27, 64, 125, 216\}$ and whose density is in $\{2, 3, 4, 5\}$; Figure 13

Table 3: Simulation Parameters

| Symbol | Meaning | Value |
|---|---|---|
| $N_{train}$ | size of seed graph | 50 |
| $\rho_{train}$ | density of seed graph | 5 |
| $N_{test}$ | sizes of tested graphs | 27, 64, 125, 216 |
| $\rho_{test}$ | densities of tested graphs in the Euclidean space | 2, 3, 4, 5 |
| $\delta_{test}$ | average node degree of tested graphs in the hyperbolic space | 1, 2, 3, 4 |
| $R$ | communication radius in the Euclidean space | 1000 |
| $-\alpha$ | negative curvature in the hyperbolic space | -0.6 |
| $\Omega = I + J$ | # of input features | 2,4 |
| $K$ | # of hidden layers | 2 |
| $N_e[]$ | # of neurons in each hidden layer | $[50\Omega, \Omega]$ |
| $\epsilon$ | margin for shortest paths prediction | 0.05 |
| $\phi$ | # of nodes for subsampling | 3 |
| $\gamma$ | discount factor | 1 |
| $IterNum_S$ | # of iterations in supervised learning | 5000 |
| $IterNum_{RL}$ | # of iterations in RL | 1000 |
| $EpiNum$ | # of episodes in RL | 20 |

plots the prediction accuracies of *Greedy Tensile* policies across graphs whose size is in {27, 64, 125, 216} and whose average node degree is in {1, 2, 3, 4}.

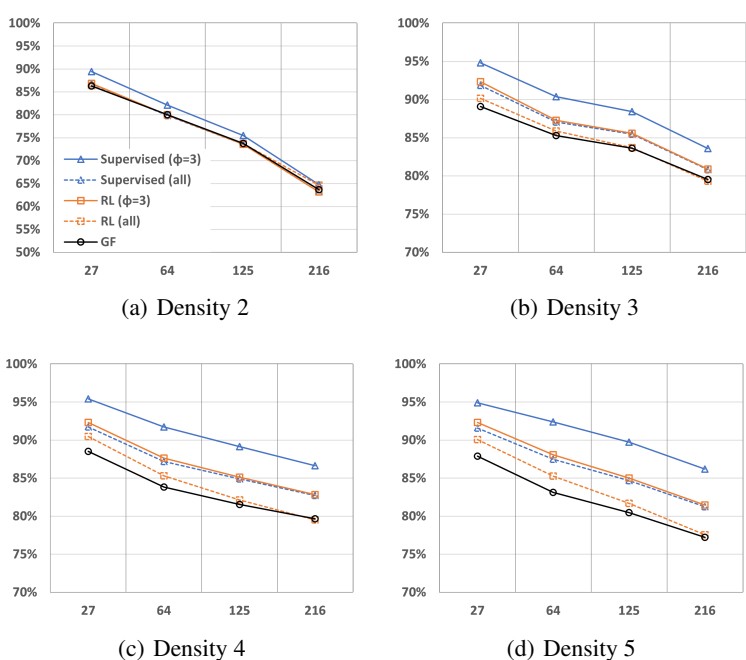

(a) Density 2    (b) Density 3

(c) Density 4    (d) Density 5

Figure 12: APNSP prediction accuracy on tested graphs of various sizes and densities in the Euclidean space for *Greedy Tensile* policies.

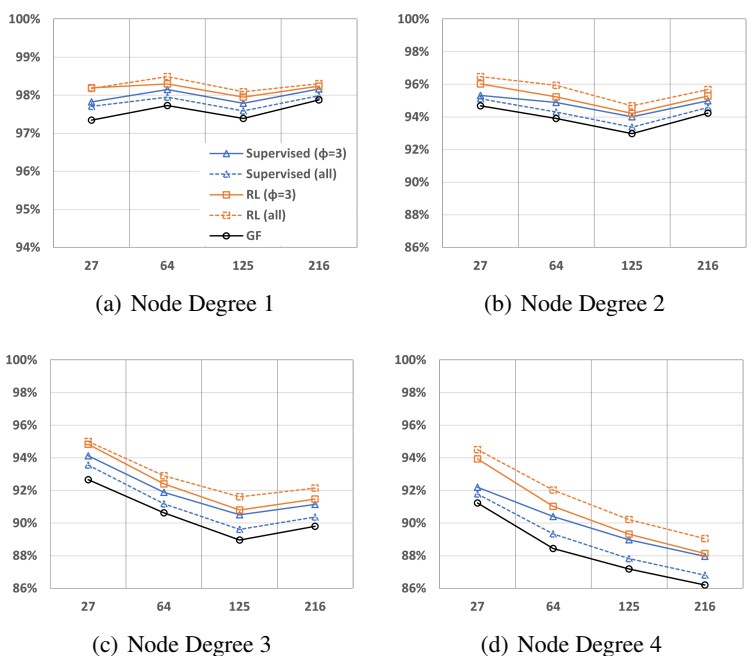

Figure 13: APNSP prediction accuracy on tested graphs of various sizes and node degrees in the hyperbolic space for *Greedy Tensile* policies.

### D.3   PERFORMANCE EVALUATION FOR THE TWO-LINEAR-ACTION POLICY

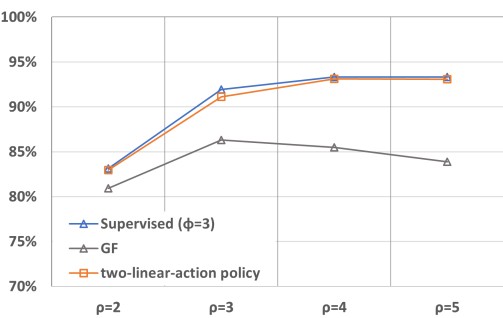

Figure 14: APNSP prediction accuracy for the two-linear-action policy in Equation 4 versus the policies of learned DNN (Supervised ($\phi = 3$)) and greedy forwarding (GF) in graphs with size 50 and density in $\{2, 3, 4, 5\}$.

The performance of the policy using Equation 4 is shown in Figure 14. The two-linear-action policy achieves an approximated accuracy to the learned DNN.

## E   SYMBOLIC INTERPRETABILITY OF LEARNED MODEL WITH DISTANCE INPUT FEATURES

Figure 15 shows that the learned DNN, using $dist(v, D), dist(u, D)$ as the input feature, is monotonically increasing as the $dist(u, D)$ decreases for a fixed $dist(v, D)$ in both Euclidean and hyperbolic spaces. Since $dist(v, D)$ stay unchanged for a fixed routing node $v$ at a given time and does not affect the ranking of $\langle m(f_s(O, D, v), f_a(O, D, u_0)), \dots, m(f_s(O, D, v), f_a(O, D, u_d)), u_i \in nbr(v)$, these models are effectively equivalently substituted for routing by a simple linear function (e.g., $-dist(u, D)$) that achieves the same performance as Greedy Forwarding.

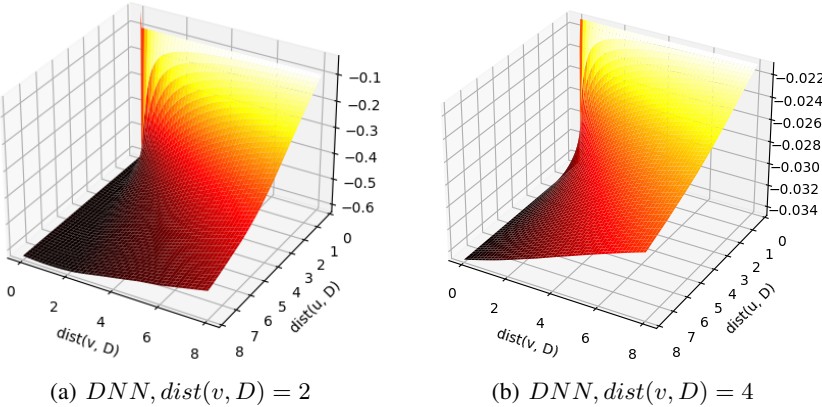

(a) $DNN, dist(v, D) = 2$    (b) $DNN, dist(v, D) = 4$

Figure 15: The shape of ranking metrics of the learned DNN using $dist(v, D), dist(u, D)$ as the input features. The x and y axes represent $dist(v, D)$ and $dist(u, D)$, and the z axis is the ranking metric for routing.

## F    COMPLEXITY OF GRAPH SUBSAMPLING

According to the results in Appendix B.1 and B.2, seed nodes must have respectively have a high $SIM_v(m, Q^*)$, denoting the ranking similarity between the ranking metric $m$ and the optimal Q-values ($Q^*$) at node $v$. The seed graph is assumed to have a sufficient number of nodes $v$ that have high $SIM_v(m, Q^*)$.

In graph subsamples selection in Section 4.1, we search the $\phi$ nodes with highest values of $SIM_v(m, Q^*)$ for graph subsampling. Typically, depth-first search (DFS) or breadth-first search (BFS) can be used for searching a graph $G = (V, E)$ to get a node list of top-$\phi$ $SIM_v(m, Q^*)$ for subsampling. The time complexity of DFS and BFS is $O(|V| + |E|)$ since the worst case of traversal time in these two algorithms should explore every node and edge. However, searching an entire graph becomes time consuming as the number of nodes increase to a considerable value. Note that in a uniform random graph with an average node degree $\delta$, the number of edges in a graph can be represented by $O(E) = O(\delta V)$.

To effectively reduce the search effort, an alternative is to search the nodes with top-$\phi$ $SIM_v(m, Q^*)$ values in a subgraph $G^{'}(D, \Lambda) = (V^{'}, E^{'}) \subseteq G$, where only nodes within $\Lambda$-hop distance to a given destination $D$ are included in the subgraph.

**Theorem 3** (Complexity of Graph Subsampling). *Search $\phi$ nodes with top-$\phi$ $SIM_v(m, Q^*)$ in a subgraph $G^{'}(D, \Lambda) = (V^{'}, E^{'}) \subseteq G$ has a time complexity of $O(\delta^\Lambda)$.*

*Proof.* Figure 8 shows the distribution of $SIM_v(m, Q^*)$ for all pairs $(v, O, D)$ in $G$ and Figure 16 plot the distribution $SIM_v(m, Q^*)$ for the pairs $(v^{'}, O^{'}, D), v^{'}, O^{'} \in V^{'}$ in the set of subgraphs, $G^{'}(D, \Lambda) = (V^{'}, E^{'}), D \in V$. Figures 8 and 16 show a similar distribution of ranking similarity. Searching nodes with top-$\phi$ $SIM_v(m, Q^*)$ from a subgraph $G^{'}(D, \Lambda)$ does not impact the efficacy of subsampling for learning to achieve high accuracy. Therefore, for a given destination $D$ in a random graph $G$ with an average node degree $\delta$, the time complexity of searching using DFS/BFS in the corresponding subgraph $G^{'}(D, \Lambda)$ is $O(\delta^\Lambda)$, a constant number independent of the graph size. □

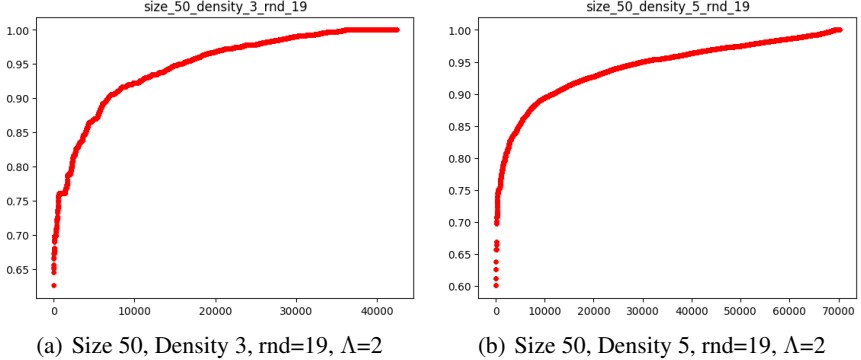

(a) Size 50, Density 3, rnd=19, $\Lambda$=2       (b) Size 50, Density 5, rnd=19, $\Lambda$=2

Figure 16: Distribution of $SIM_v(m, Q^*)$ in the set of subgraph $G^{'}(D, \Lambda)$ of $G = (V, E)$ in Figure 8. Each subgraph $G^{'}(D, \Lambda)$ includes nodes only within $\Lambda$-hop distance to the given destination $D \in V$.

