# OpenReview forum: "Learning from A Single Graph is All You Need for Near-Shortest Path Routing"
_ICLR.cc/2024/Conference — Submitted to ICLR 2024_

### Official Review · Reviewer_HSXi · 2023-10-24

**Soundness:** 2 fair
**Presentation:** 2 fair
**Contribution:** 2 fair
**Rating:** 3
**Confidence:** 4

**Summary:**

This paper addresses the All-Pairs Near-Shortest Path (APNSP) problem using the Markov Decision Process (MDP) framework and proposes a DNN-based approach to learning the local forwarding policy by predicting the Q-values. The neural network model takes the state features and action features as input, which include the distance information of the current state (node) and the next state (one neighbor of the current node). The models are trained based on supervised learning, where the samples are collected from a seed graph. In addition, the authors provide theory results to demonstrate the generalization properties of the model. Experiments have been shown to evaluate the scalability and generalizability of the proposed approach.

**Strengths:**

- S1: This paper provides a practical MDP framework for the APNSP problem.
- S2: The empirical results of the proposed approach show comparable performance to the baseline greedy forwarding approach.

**Weaknesses:**

- W1: The proposed approach is reasonable but lacks novelty. Using Q-learning as a heuristic to solve optimization problems is straightforward and somehow trivial. Regarding the APNSP problem, similar approaches have been previously proposed, e.g.,
  -  Wu, Yaoxin, Wen Song, Zhiguang Cao, Jie Zhang, and Andrew Lim. "Learning improvement heuristics for solving routing problems." IEEE transactions on neural networks and learning systems 33, no. 9 (2021): 5057-5069.
  - Bi, Jieyi, Yining Ma, Jiahai Wang, Zhiguang Cao, Jinbiao Chen, Yuan Sun, and Yeow Meng Chee. "Learning generalizable models for vehicle routing problems via knowledge distillation." Advances in Neural Information Processing Systems 35 (2022): 31226-31238.


- W2: The theoretical results of the generalizability of the proposed approach are not warranted. The concept of “learnable” mentioned in Theorems 0, 1, and 2 needs formal treatments – for example, sample complexity, PAC-learnability, and regret bound.
  - W2-1: The RankPre property and optimal ranking seem to be the core concepts of the theorem, but the relationship between these two concepts and the model’s generalizability power is unclear. First, the RankPre seems to be the property of the raking metric m. It looks weird because the property's content is like a conclusion. Second, regarding the optimal ranking, the authors require the monotonically increasing order of the ranking metric for the neighbors of a node in the graph. However, there are no arguments showing the reason for this requirement and its relationship with the generalizability results.
  - W2-2: it is confusing to me about  “a learnable DNN”. The authors claim that there exists a learnable DNN that can achieve optimal ranking without any further description. For example, what is the hypothesis space? How to train the model, and what is the sample complexity?
  - W2-3: In the proof of theorem 0, the authors claim that the DNN can achieve “optimal ranking” by learning a group of weights for each input feature that matches those used in the ranking metrics. From my understanding, this means a zero empirical risk which does not have any further discussions.
  - W2-4:  It is unclear how closely the implementable algorithm relates to the theoretical results.  The proposed approach does not seem to be scalable. It is not clear about “a subset of training samples”.  Even from an intuitive perspective, according to the description, if this collected subset sample is limited to only a small portion of the graph (including the origin and destination), how can the model learned from here be applied to all graphs?

- W3: There’s no experimental comparison with other state-of-the-art approaches for the All-Pairs Near-Shortest Path (APNSP) problem.

- Minor comments:
  - In section 2.2, when defining the Q-value, it is unclear about the notation L and t.
  - The authors explain the idea behind designing the input features in Proposition 1, wherein a ranking metric using the designed distance input features following a specific format that meets the RankPres property. It is not straightforward to me why this format would satisfy the property. Detailed proof would be helpful.
  - The shapes in Figure 3 are hard to recognize.

**Questions:**

What does it mean, mathematically, by "learnable" in the theorems?

---

### Official Review · Reviewer_D6sq · 2023-10-30

**Soundness:** 2 fair
**Presentation:** 3 good
**Contribution:** 1 poor
**Rating:** 1
**Confidence:** 4

**Summary:**

This paper looks at the (near-)shortest path routing problem on geometric random graphs in Euclidean and hyperbolic metric spaces. Each instance of the problem consists of a graph and a pair of source and destination nodes. There are two settings, one where the input feature for each node is simply the (Euclidean/hyperbolic) distance to the destination and one where each node additionally receives its stretch factor (essentially how far the node lies off of the straight line distance between the source and destination nodes) as input.
The authors train a neural network to calculate a score independently for each neighbour of a node, so that the highest score (Q-value) neighbour can be chosen as the next location towards the destination. The input for the scoring network is the node itself and one of its neighbours. By repeating the scoring for each neighbour, a step can be taken, and by repeating these steps multiple times, a path can be formed from the source to the destination node.
The authors show that their neural network with just the distance to the destination as input is able to learn the greedy policy of always choosing the neighbour that is nearest to the destination. They also show that their neural network can outperform greedy when the stretch factor is also provided. Their network is able to learn effective policies with very little training data.

**Strengths:**

The paper is easy to follow. Routing is generally an interesting topic, though a bit narrow.

**Weaknesses:**

The problem is framed in a way that the neural network has very little learning to do. In the simpler case the network essentially just has to learn to put a minus sign in front of a neighbour’s distance to the destination. In this way the node with the highest score will correspond to the neighbour with the lowest distance to destination and will be selected as per the greedy heuristic. Even in the slightly more complex setting, the network learns a simple policy - as revealed also by the authors in Figure 3 and Equation (4).

There is no interesting contribution in terms of the wider setting of neural algorithm learning. There are many other papers that replace a heuristic step in an algorithm with a neural network, so this cannot be considered a novel approach.

The problem setting is very narrow and easily (approximately) solved with very simple (and very efficient) heuristics. The authors do not show any significant advantage over using a neural network to replace these simple heuristics, neither in terms of performance, nor in terms of efficiency.

Barring a contribution in terms of performance or efficiency, I would expect new results in terms of what a neural network is able to do, but learning a simple linear relation is nothing surprising, even if very little data is required. Planning multiple steps ahead would already be much more interesting and more complex for this problem. Of course this would require giving the whole graph as an input to the neural network, rather than just a pair of nodes.

The main selling point of this paper seems to be how little data is required to learn an algorithm that generalises effectively. But in the context of what the neural network has to learn here, this is not particularly impressive.

Little effort has been made to push the model to its limits. One could include a set of “difficult” graphs, where the greedy strategy fails.

Algorithmically this problem both settings are well understood since 20 years. It's not clear why we need a learning approach in the first place.

**Questions:**

The major questions are in the weaknesses. Here are some minor questions:
1. In Section 2.3, the distance to destination is presumably the Euclidean/hyperbolic distance and not the graph distance, but this is not clearly specified.
2. In Theorem 1 we assume that a (linear) ranking metric that satisfies RankPres exists (i.e. gives an optimal ranking of neighbouring nodes) and claim that it is therefore learnable. But unless I am missing something, the theorem then essentially just says we can learn a linear function with a neural network. Calling this a Theorem seems to be an overstatement. And surely the interesting cases would be when the assumption does not hold.
3. Where is the proof for Proposition 1?
4. Why are the page numbers in roman numerals?
5. Footnote 1 on page v essentially describes “monotonically non-decreasing”, perhaps you want to use this term instead.
6. In Figure 2, what epsilon value is used?
7. Throughout the paper, accuracy is reported, but what about other metrics? For example, the average relative cost increase versus the optimal solution would be interesting.
8. Where are the results that confirm the claim on page vii “that the performance of all the learned policies exactly match the prediction accuracy of Greedy Forwarding”?
9. Do you have a citation to back up the claim on page viii that “GF was believed to work close to the optimal routing”? There are several theoretical works that go beyond greedy forwarding. Moreover, these works could even be used as baselines.

---

> ### Author Response · Authors · 2023-11-21
>
> Q: In Figure 2, what epsilon value is used?
>
> A: We list the simulation parameters in Table 3 in the appendix. The $\epsilon$ value (i.e., the margin for shortest paths prediction) is set to 0.05.
>
> Q: Do you have a citation to back up the claim on page viii that “GF was believed to work close to the optimal routing”?
>
> A: The following citation [3] states that Greedy Forwarding achieves nearly optimal routing in hyperbolic space.
>
> [3] Fragkiskos Papadopoulos, Dmitri Krioukov, Mari´an Bogun´a, and Amin Vahdat. "Greedy forwarding in dynamic scale-free networks embedded in hyperbolic metric spaces." In IEEE INFOCOM, pp. 1–9, 2010.

---

### Official Review · Reviewer_JN8R · 2023-10-31

**Soundness:** 3 good
**Presentation:** 3 good
**Contribution:** 1 poor
**Rating:** 3
**Confidence:** 3

**Summary:**

The authors use neural networks to solve the problem of finding (approximately) shortest paths in geometric graphs using only local information.

They consider graphs obtained by sampling a number of points in Euclidean or hyperbolic space, and connecting with an edge any two points whose distance is smaller than a fixed threshold. In such graphs, they consider routing algorithms that given current node u, and knowing that we are going from s to t, pick the next node v from neighbors of u based only on information about s, t, and u (and not about the whole graph). A classic heuristic of this kind, called Greedy Forwarding, simply picks the neighbour that minimizes the geometric distance to target t.

The authors propose a more elaborate approach for routing. They associate with each node v two features that depend on the source and target: 1) geometric distance from target d(v,t) and 2) stretch (d(s,v)+d(v,t))/d(s,t). Then, they train a neural network that takes as input features of two neighboring nodes u, v and outputs an approximation of the so-called Q-value. The Q-value is supposed to be high if it is a good routing decision to go to node v when being at node u, and in practice it is simply the negative length of the shortest path from v to the target. Then, to perform the actual routing, one has to evaluate the model on all neighbors of the current node and pick the one that gave the highest output value.

The network is trained using either supervised learning or RL, using only a small sample of nodes from a single graph. Both approaches generalize well and beat the Greedy Forwarding method. The metric used for evaluating routing approaches is the percentage of node pairs for which the evaluated approach finds a path with length within a multiplicative constant from the true shortest path length. I have not found information on what the constant is.

The authors also analyze what the network has learned. If only the distance feature is used, the network learns to mimic the Greedy Forwarding algorithm (which is nice, but not very surprising, see, e.g., "A new dog learns old tricks (...)" ICLR'19 paper). When both features are used, the network's behavior can be approximated with a piecewise linear function with only 2 pieces.

**Strengths:**

The presented approach improves over the standard Greedy Forwarding algorithm while still being reasonably simple to implement.

The paper is well written and easy to read.

**Weaknesses:**

The comparison with prior work is insufficient. The only benchmark used is Greedy Forwarding, which uses only one feature (distance from target). It is not clear if the presented improvement comes from using more features or from a more elaborate method to use these features. More importantly, it is not clear how other (previously known and possibly simpler) heuristics compare to the proposed approach.

I do not like the fact that the authors start with a neural network without checking first a simpler model (e.g. linear regression).

Even though the authors keep using the phrase "deep neural networks", they only use networks with two hidden layers.

Even though this is primarily an experimental paper, and the experiments described do not seem to require any specialised hardware nor proprietary datasets, the authors do not provide their source code nor anything else that would make reproducing their results easier (or at least I haven't found any such thing).

**Questions:**

Have you tried simpler models, say linear regression, before using neural networks?

Could you provide experimental comparison with some prior works that also use stretch factor?

What do you mean by "class of all graphs whose nodes are uniformly distributed" (page 2)? Do you mean class of distributions over graphs?

What value of zeta(O,D) do you use in experiments?

---

> ### Author Response · Authors · 2023-11-21
>
> Q: Have you tried simpler models, say linear regression, before using neural networks?
>
> A: For the input features based on only distance, using linear regression also learns a ranking metric that matches the greedy forwarding routing. However, for the input features based on both distance and stretch factor, using linear regression learns a ranking metric that underperforms the one learned using neural networks in terms of the APNSP prediction accuracy.
>
> Q What value of $\zeta(O,D)$ do you use in experiments?
>
> A: Since $\zeta(O,D)$ denotes the path stretch of the endpoints (i.e., $\frac{d_{sp}(O,D)}{d_{e}(O,D)}$), it varies for different (O, D) pairs across different random graphs.

---

### Official Review · Reviewer_zGzT · 2023-11-04

**Soundness:** 2 fair
**Presentation:** 2 fair
**Contribution:** 3 good
**Rating:** 6
**Confidence:** 3

**Summary:**

This work presents a novel idea of deriving local routing policies using MDP formulation and small set of random graphs in Euclidean and hyperbolic metric spaces. The idea is supported by theories and their proof. The local routing policy can be trained by two methods: 1) supervised learning with shortest path distance known or 2) deep reinforcement learning. Experimental results show that the trained policy outperforms the greedy routing algorithm.

**Strengths:**

The paper tackles a critical routing problem in computer network, either in wireless or wired setting. The work presents the fundamental graph routing problem using generalized graphs and argues that one can train a DNN that has a local optimal policy with sampled seed graphs.

**Weaknesses:**

The work presents theories and their proof. However, the overall writing is not easy to follow. In addition, the experimental results are limited. The work only considers 20 graphs in evaluation of the policies.

**Questions:**

Can the proposed method used in graphs where the number of neighboring nodes can be different for every node? From the example and the formulation, it seems that the number of neighboring nodes is constant. If the number can vary, how do you setup your DNN and RL framework?

---

> ### Author Response · Authors · 2023-11-21
>
> Q: Can the proposed method used in graphs where the number of neighboring nodes can be different for every node? From the example and the formulation, it seems that the number of neighboring nodes is constant. If the number can vary, how do you setup your DNN and RL framework?
>
> A: The proposed method takes $f_s(O,D,v)$, $f_a(O,D,u)$ as the input to predict the $Q$ value for an edge $(v, u)$. It is applied to each of the neighbors of node v, regardless of the degree of $v$, to predict the respective $Q$ values for all of its neighbors and to thus select a neighbor with the highest predicted $Q$ value as the next forwarder. Thus, it suffices for graphs whose nodes have different degrees.

---

> > ### Comment · Reviewer_zGzT · 2023-11-22
> >
> > Can you please elaborate the implementation details if the work can be applied to graphs whose nodes have different degrees?

---

> ### Author Response · Authors · 2023-11-22
>
> Let us assume we can learn a model with a given training sample set from a seed graph $G^*$ (size= 50, density=2) whose nodes have a number of neighbors varying from 1 to 3. Let the data sample set be derived from node $v$ and its three neighbors $u_1$, $u_2$, and $u_3$. Thus, the data sample set includes respectively three X and Y vectors: $X = \{\langle [f_{s}(O, D, v),f_{a}(O, D, u_1)] , [f_{s}(O, D, v),f_{a}(O, D, u_2)], [f_{s}(O, D, v),f_{a}(O, D, u_3)] \rangle\}$; $Y= {\langle [Q^*(v,u_1)],  [Q^*(v,u_2)], [Q^*(v,u_3)] \rangle\}$.
>
> Note that $Q^*(v, u)$ denotes the optimal $Q$ value (i.e., the negative length of shortest path from $v$ to $D$ passing through $u$).
>
> Again, let us assume model $m'$ is learned with the data sample set to predict a Q value for a given a routing node $v$ and its neighbor $u$ associated with a origin-destination pair $(O, D)$. Our work enables to test the learned model in a graph with different size and density. Let the test graph $G'$ (size= 100, density=4) have node degrees varying from 1 to 5. For a routing node $v'$ in $G'$ and its five neighbors $u_i'$, $i = 1 ... 5$, we feed $[f_{s}(O', D', v'),f_{a}(O', D', u_i')]$ into the model $m'$ to output the single predicted $Q$ value for each $(v', u_i')$ pair, respectively. After the five times prediction for all $v'$'s neighbors $u_1'$, $u_2'$, $u_3'$, $u_4'$, and $u_5'$, we will receive five predicted $Q$ values. The neighbor with the highest predicted $Q$ value will be selected as the next forwarder.

---

### Official Review · Reviewer_KPLY · 2023-11-05

**Soundness:** 1 poor
**Presentation:** 2 fair
**Contribution:** 3 good
**Rating:** 3
**Confidence:** 3

**Summary:**

This paper studies the problem of finding approximate all-pair shortest paths for 2D Euclidean and Hyperbolic random graphs. Specifically, the shortest paths should be returned in a localized way, such that given source and destination s and t and an intermediate vertex v, it returns the next “hop” from v towards the nearly-shortest path to the destination.

The motivation to consider the two types of random graphs is that they are good models for capturing sensor networks and social networks.

At a high level, the proposed method is for every vertex v find a ranking of the neighbors of v using DNN. The crucial claim is that, if there exists a good enough *linear* ranking metric, then simply learning on a single seed graph would generalize to all graphs. Then the next step is to construct such a ranking metric for the two special graph families (i.e., Euclidean and Hyperbolic). The step of finding a ranking metric is via heuristic methods that are tested in the experiments.

**Strengths:**

The problem is well-motivated.

The method of learning from a single graph is an interesting proposal, and figuring out how/why this works is an interesting research problem.

The algorithms are designed in a systematic way through a framework (e.g., defining and analyzing RankPres ranking metrics) instead of testing a collection of ad hoc heuristics.

**Weaknesses:**

The technical presentation is mathematically informal, and the statement of Theorems (and even definitions) is very unclear. This makes it extremely difficult to justify the correctness of the proofs/claims, which in turn questions the soundness of the proposed “learning from a single graph” method. I listed several concrete gaps in the proofs in the next “Questions” section.

**Questions:**

1. Page 3, you mention that V is inside a square of side length \sqrt{n R^2 / \rho} — This bound makes sense to me only when you restrict the random process in a finite R^2 area, instead of the entire Euclidean plane. However, you did not explicitly mention that the points are only drawn from R^2 area.

2. In Section 2.1, the paragraph of “The APNSP Problem”:

- How is v quantified In the mapping \pi(O, D, v)? Can v be arbitrary? What if v is far from any near-shortest path between O and D?

- You mentioned “\xi(O, D) (1 + \epsilon)” is the user-specified factor — I think that only \epsilon is what the user can specify, since \xi(O, D) even depends on the algorithm which is designed after the user specifies the parameters.

- What’s the variables of the “max” in equation (1)? Over all \pi? And what about G — I think that G is randomized, so it does not make sense to take the max over G; instead, you may want to define (1) as something like max_\pi E_G[Accoracy_{G, \pi}].

3. Page 4, second paragraph of section 3:

- You said f_s and f_a maps from V, but then you used f_s(O, D, v) and f_a(O, D, u), whose parameters are not from V.

- What is Q-value, did you define it exactly? At least you should provide some reference. This is important since your Theorems depend on Q-value, so the proof depends on the exact definition of it.

4. Theorem 1:

- How small can the V’ be? I think V’ = V seems to always work, and not restricting the size of V’ makes this claim useless (?)

- Can you give a formal definition of the meaning of “learnable” in this context?

- From the proof of Theorem 1, it seems you want to say as long as the RankPres property holds for all v \in V, and you just train H using any one v \in V as in Theorem 0, then the H value on any other v’ \in V also preserves the ranking. This is a strong claim, and I don’t see why this is true. The fact that Theorem 0 can yield a good H for a given v does not mean the *same* H can work for every other point v’ — why couldn’t it be that you apply Theorem 0 again on v’ and you get a different H’?

5. Theorem 2:

- Notice that Theorem 1 is applicable on one (fixed) graph G. But in the proof of Theorem 2, it seems you want to say you can apply Theorem 1 on an arbitrary graph, then wishes that it preserves the learnability property for *all* graphs simultaneously. This is a similar issue as in the proof of Theorem 1.

6. In the paragraph immediately below Theorem 1 (and also a paragraph immediately below Theorem 2), you discussed the case when the RankPres property is not satisfied for all nodes, but for most of them, then “with high probability” things can still work. I’m not sure — what do you mean by “with high probability”? What’s the randomness in this context? Can you give some formal justification?


7. In section 4.1, you mention you need to choose the seed graph carefully. However, in Theorem 2, it does not seem to have any restriction on the seed graph, which means any graph can make Theorem 2 hold. Then what’s the point of discussing how to select a seed graph here? In what sense could it help you?

---

> ### Author Response · Authors · 2023-11-21
>
> Q: In section 4.1, you mention you need to choose the seed graph carefully. However, in Theorem 2, it does not seem to have any restriction on the seed graph, which means any graph can make Theorem 2 hold. Then what’s the point of discussing how to select a seed graph here? In what sense could it help you?
>
> A: In Appendix B, we adopt Discounted Cumulative Gain (DCG) to formally define the ranking similarity and choose the seed graph that achieves the highest average ranking similarity across nodes. As evidenced by Figures 4-9, a good choice of seed graph is likely to exist in a set of graphs with modest size (e.g., 50) and high density (e.g., 5) in the Euclidean space and with high average node degree (e.g., 4) in the hyperbolic space.

---

### Official Review · Reviewer_w67p · 2023-11-07

**Soundness:** 1 poor
**Presentation:** 3 good
**Contribution:** 2 fair
**Rating:** 3
**Confidence:** 3

**Summary:**

The authors consider the problem of learning a local routing policy in a graph. This policy uses only information available at a node and its surrounding nodes. The authors present some results regarding conditions under which it is possible to learn the optimal ranking policy given a set of features. The authors formulate the routing problem as a deterministic Markov decision process, and propose a supervised learning procedure that estimates the optimal Q-values of a seed graph. The authors provide some computational experiments comparing the effectiveness of this procedure to a reinforcement learning approach and a greedy forwarding approach.

**Strengths:**

The paper is, for the most part, clearly written and well-organized. The problem of learning local routing policies in a graph is fairly interesting.

**Weaknesses:**

Theorems 0 to 2 have extremely strong assumptions. In particular, the RankPres property is a very strong property. These results could be strengthened greatly by first identifying some weaker property, and showing some weaker result under this property. For example, you could perhaps show that there is some probability that the RankPres approximately holds, and then have some result that shows that there is some policy with some bounds on the error that can be obtained in such a case. For this reason, Theorem 0 is nearly trivial.

The authors have not clearly defined the phrase "learnable", which plays a key role in Theorems 0, 1, and 2, but I believe that Theorem 1 is false for any reasonable interpretation of this phrase. For example, one could define a graph wherein there is a node $v$ with a unique neighbor $u$. Then, the training sample for the singleton set ${v}$ would consist of a singleton set $\{(X_v, Y_v) \}$. This isn't enough to characterize a relationship between the local features and the Q-value function.

The authors call "Proposition 1" a proposition, but provide only numerical experiments that they claim validate this proposition. Further, these numerical experiments do not validate this proposition at all. The proposition claims that a ranking metric that satisfies the Rank Pres property exists for almost all nodes in almost all graphs G. The numerical experiments show that a particular ranking metric approximately satisfies this property for a large proportion of graphs. This leaves a large gap. I am almost certain that this proposition is false as written, unless an atypical meaning is given to the phrase "almost all nodes in almost all graphs G". It seems that Euclidean graphs that do not admit a RankPres metric based on the given features should occur with non-zero probability. I suspect that this proposition holds asymptotically, both as $n \to \infty$ and as $\rho \to \infty$, but this remains to be shown.

The choice of seed graph seems like it should be quite important, but I could not find any details on how this graph was selected. Is it just randomly generated with the parameters listed in Table 3?

The accuracy metric used by the authors is somewhat strange. It is okay to show some results using this metric, but it would be better to additionally show results that are instead based on a more normal measure of performance, such as the average/median ratio between the path achieved and the shortest path.

**Questions:**

In theory, as density of the graph increase, the greedy policy should become close to optimal. However, this is not reflected in Fig (a) of the computational results. Can you explain the discrepancy?

How is the seed graph selected?

A much more natural accuracy metric would be to define $\eta(O,D) = \begin{cases}1 \textup{ if } \frac{d_p(O,D)}{d_{sp}(O,D)} \leq 1+\epsilon, \\\\ 0 \textup{ otherwise}. \end{cases}$

Why did the authors additionally include the factor $\zeta(O,D)$ in their accuracy metric?

The authors seem to assume that a linear model in the features would be sufficient to produce an optimal policy. If this is the case, why bother with an entire deep neural network, rather than a simpler model (such as a linear model)?

---

> ### Author Response · Authors · 2023-11-21
>
> Q: Why did the authors additionally include the factor ζ(O,D) in their accuracy metric?
>
> A: The use of path stretch ζ(O,D) in Equation 2 is to normalize the margin for shortest paths prediction (given ϵ is constant).  The results in [2] show that the variance of path stretch becomes more significant as the network density ρ decreases. In sparse graphs (e.g., ρ=1.4, 2), the path stretch can vary in [1.0, 10.0] since some of the shortest paths is prone to be found around the holes. However, the path stretch in dense graphs (e.g., ρ=4, 5) is likely to vary only within the range of [1.0, 3.5]. Thus, we exploit ζ(O,D) to mitigate the gap between sparse and dense networks for APNSP prediction .
>
> [2] Chen, Yung-Fu, Kenneth W. Parker, and Anish Arora. "QF-Geo: Capacity Aware Geographic Routing using Bounded Regions of Wireless Meshes." arXiv preprint arXiv:2305.05718 (2023).

---

### Author Response · Authors · 2023-11-21
**Response to common questions about our theoretical results**

We thank all the reviewers for the insightful comments. Below we first clarify several common questions about our theoretical results and then provide a detailed response to each reviewer.

[RankPres property]

We agree with the reviewers that RankPres property is a strong assumption in theory. However, we would like to point out that our theory is mainly developed to explain the superior generalization performance of the learned neural network model in practice. In fact, in our algorithm design and also stated in Appendix B, we consider the ranking metric based on distance for greedy forwarding and distance and node stretch for our algorithm. By comparing the order of $m$ and the order of $Q^*$ based on DCG, we show in Figure 4-11 that RankPres property indeed holds for most nodes and most graphs across different size and density/average node degree, which is the reason that the learned neural network can generalize well across different graphs. Therefore, we seek to use our theoretical results to explain why generalization is possible when RankPres property holds. But in practice, good generalization performance can still be achieved even if this property only holds for most nodes and graphs, as demonstrated in our experimental results

[Interpretation of “learnable”]

Theorems 0, 1, and 2 assume (perfect) correlation of the local ranking metric $m: f_{s} \times f_{a} \rightarrow \mathbb{R} $ and the global ranking metric $Q^*$ across nodes and graphs. In Theorem 1 (respectively, 2), our objective is to learn a function, $m’: f_{s} \times f_{a} \rightarrow Q$, that maximizes the number of nodes satisfying the RankPres property in a graph (respectively, across random graphs of some sort). Note that $m$ and $m’$ achieve the same ranking order across nodes and graphs.

[Interpretation and correctness of Theorem 1 and 2]

In Theorem 1, if there exists $m()$ that satisfies the RankPres property for all nodes, then some function $m’: f_{s} \times f_{a} \rightarrow Q$ can be learned with a subset of nodes $V'$, where $V^′ \subseteq V$ and $|V^′| \geq 1$. Note that it is sufficient to learn $m’$ with the training samples derived at least from one node $v$ with the highest degree (degree($v$)>1) and that satisfies the RankPres property for all nodes. Thus both $m$ and $m’$ can serve as an optimal ranking policy. The proof is as follows.

[Monotonicity of the learned function]

RankPres implies that given $Q^*$ values for any node $v$ and its neighbors $u_1$, $u_2$, … to have an increasing order, $m(f_{s}(O, D, v), f_{a}(O, D, u_1))$, $m(f_{s}(O, D, v), f_{a}(O, D, u_2))$, … are in the same increasing order. Let us learn a function $m’: f_{s} \times f_{a} \rightarrow Q$ for some nodes $v’ \in V’$ such that $m’$ is monotonic non-decreasing, which is feasible cf. [1], for nodes $v’$ and its neighbors with respect to their increasing $Q^*$ values.  Now test the function $m’$ on some nodes $v’’ \in V \setminus V’$ and its neighbors $u1’’$, $u2’’$, …. Without loss of generality, let us assume that $m(f_{s}(O, D, v’’), f_{a}(O, D, u_1’’))$, $m(f_{s}(O, D, v’’), f_{a}(O, D, u_2’’))$, … are in monotonic non-decreasing order. Since $m’$ is monotonic non-decreasing, for this neighbor ordering, by definition their $Q$ values (i.e., the output of the function $m’$ will also preserve the same order. Hence, both $m()$ and $m’()$ preserve the RankPres property for all the nodes.

[1] Sill, Joseph. "Monotonic networks." Advances in Neural Information Processing Systems 10 (1997).

In Theorem 2, if there exists $m()$ that satisfies the RankPres property for the nodes in all graphs, then a function $m’: f_{s} \times f_{a} \rightarrow Q$ can be learned by using training samples from one or more nodes in one or more chosen seed graph(s) $G^*$. It is sufficient to learn $m’()$ with the training samples derived at least from a node with the highest degree in the seed graph $G^*$. Both $m()$ and $m’()$ can serve as the optimal ranking policy.

[Linearity of the ranking metric]

We thank the reviewers to bring up the assumption on linearity of the ranking metric. In fact, the ranking metric $m()$ can be linear or nonlinear. Our results hold as long as $m()$ is monotonic and we will remove the assumption on linear $m()$.  So instead of using linear regression, we apply neural networks to learn $m’()$ to maximize the number of nodes that achieve optimal routing for a given pair $(O, D)$ by using $m’()$. This is illustrated by Figure 3 which shows the nonlinearity of $m’()$ learned for the GreedyTensile policy. Besides, the learned ranking metric using neural networks outperforms the one using linear regression in terms of the APNSP prediction accuracy.

---

### Meta-Review · Area_Chair_3KhC · 2023-12-05

**Metareview:**

The authors study the problem of learning local routing strategies in a graph. In particular, they first reformulate the problem as a deterministic Markov decision process then provide a learning algorithm for it. They also provide theoretical and experimental results showing the effectiveness of their algorithm.

Overall, the paper contains some interesting ideas but it is not ready for publication at this stage. More specifically, the reviewers highlighted the following weaknesses:

- The assumption in the theoretical results are too strong and a bit hard to parse

- The presentation of the mathematical results should be improved

- The comparison with previous work is not particularly satisfying

**Justification For Why Not Higher Score:**

The paper is not ready for publications, reasons for rejection are listed above

**Justification For Why Not Lower Score:**

N/A

---

### Decision · Program_Chairs · 2024-01-16

Reject